

# An efficient two-layer landslide-tsunami numerical model: effects of momentum transfer validated with physical experiments of waves generated by granular landslides

Martin Franz[1], Michel Jaboyedoff[1], Ryan P. Mulligan[2], Yury Podladchikov[1], W. Andy Take[2]

[1]Institute of Earth Sciences, University of Lausanne, Lausanne, CH-1015, Switzerland
[2]Department of Civil Engineering, Queen's University, Kingston, ON K7L 3N6, Ontario, Canada

*Correspondence to*: Martin Franz (mart.franz@gmail.com)

**Abstract.** A landslide-generated tsunami is a complex phenomenon that involves landslide dynamics, wave dynamics and their interaction. Numerous lives and infrastructures around the world are threatened by these events.

Predictive numerical models are a suitable tool to assess this natural hazard. However, the complexity of this phenomenon causes such models to be either computationally inefficient or unable to handle the overall process. Our model, which is based on shallow water equations, is developed to address these two problems. In our model, the two materials are treated as two different layers, and their interaction is resolved by momentum transfer inspired by elastic collision principles.

The goal of this study is to demonstrate the validity of our model through benchmark tests based on physical experiments

performed by Miller et al. (2017). A dry case is reproduced to validate the behaviour of the landslide propagation model using different rheological laws and to determine which law performs the best. In addition, a wet case is reproduced to investigate the influence of different still water levels on both the landslide deposit and the generated waves.

The numerical results are in good agreement with the physical experiments, thereby confirming the validity of our model, particularly concerning the novel momentum transfer approach.

## 1 Introduction

A landslide-generated tsunami is a complex phenomenon that involves landslide dynamics, interactions between the landslide mass and a water body, propagation of a wave and its spread on the shore. A landslide could be either submarine or subaerial. Regions that combine steep slopes and water bodies are the most susceptible to landslide-generated tsunamis. For instance, fjords (*Åknes*: Ganerød et al., 2008; Harbitz et al., 2014; Lacasse et al., 2008, *Lituya Bay*: Fritz et al., 2009,

Slingerland and Voight, 1979, Weiss et al., 2009), volcanos in marine environments (*Stromboli*: Tinti et al., 2008, *Réunion Island*: Kelfoun et al., 2010), and regions such as lakes and reservoirs in mountainous areas are prone to this phenomenon (*Chehalis Lake*: Roberts et al., 2013, *Vajont*: Bosa and Petti, 2011; Slingerland and Voight, 1979; Ward and Day, 2011; Kafle et al., 2019; *Lake Geneva*: Kremer et al., 2012, 2014, *Lake Lucerne*: Schnellmann et al., 2006). On plains, landslide-generated tsunamis are also encountered when particular conditions exist, such as quick clays or glacio-fluvial deposit slopes



(*Rissa*: L'Heureux et al., 2012, *Nicolet Landslide*: Jaboyedoff et al., 2009; Franz et al., 2015, *Verbois reservoir*: Franz et al., 2016).

Landslide-generated tsunamis severely threaten lives and infrastructures, as evidenced in Papua New Guinea in 1998, where a submarine landslide killed 2200 people (Tappin et al., 2008). To assess this hazard, predictive models must be used. These models can be separated into 3 different types: 1) empirical equations from physical models (Enet and Grilli, 2007; Heller et

al., 2009., Fritz et al., 2004, Miller et al., 2017, Kamphuis and Bowering, 1970; Slingerland and Voight, 1979), 2) physical models reproducing site-specific setups (*Åknes*: Harbitz et al., 2014, *Vajont*: Ghetti, 1962 in: Ghirotti et al. 2013), and 3) numerical models. Numerical models can be governed by different sets of equations, such as smoothed-particle hydrodynamics (Heller et al., 2016; Wei et al., 2015), shallow water equations (Simpson and Castelltort, 2006; Harbitz et al., 2014; Franz et al. 2013, 2015, 2016; Kelfoun 2011; Kelfoun et al., 2010; Mandli, 2013; Tinit and Tonini, 2013; Tinti et al.

2008), and Boussinesq equations (Harbitz et al., 2014; Løvholt et al., 2013). Furthermore, numerical models can be simulated in full 3D (Crosta et al., 2013), with a less classical hybrid approach (Xiao et al., 2015; Ward and Day, 2011), and with advanced multi-phase approach (Pudasaini, 2014; Kafle et al., 2019).

The assessment of natural hazards requires predictive numerical models that are able to sufficiently reproduce the studied phenomenon while being efficient in terms of computational resources. The ease of implementation (few selected

parameters) is also a great advantage. Models based on shallow water equations are a good compromise from this point of view (Franz et al. 2013). However, few of these models assess the whole phenomenon, i.e., simulating both the landslide propagation and the tsunami. To perform such an assessment, the model must handle complex behaviour, in particular sliding mass/water momentum transfer, wet-dry transition, and flooding.

Kelfoun et al. 2010 presented the *Volcflow* model, which has the ability to handle such behaviour. In this model, the

momentum transfer is performed by a set of drag-like equations modified from the methodology reported by Tinti et al. (2006). Their approach is an elegant way to solve this type of problem; however, this method also relies on complex assumptions linked with the free-surface nature of the model. Xiao et al. (2015) simulated momentum transfer through a so-called "drag-along" mechanism. This approach is relevant but requires the implementation of coefficients that are subject to interpretation. The two-phase mass flow model proposed by Pudasaini (2012) automatically simulates landslides and

tsunamis within a single framework (Pudasaini, 2014; Kafle et al., 2019).

The numerical model we propose in this study is a two-layer model that combines a landslide propagation model and a tsunami model. The proposed numerical model is based on shallow water equations in an Eulerian specification of the flow field. The transfer of momentum between the two layers is performed by assuming a perfectly elastic collision. Although this method is obviously wrong from a physical perspective, it can be an option to compute the momentum transfer between two

materials in which each cell has a certain velocity and mass. This approach has the advantage of containing a limited number of coefficients to be implemented by the operator. The "lift-up" mechanism (i.e., the change in height of the water level due to bed rise) also contributes to the wave generation.


The aim of this study is to test the whole model (i.e., landslide and tsunami) and more specifically to examine the transfer of momentum between the two materials. Miller et al. (2017) provided a relevant benchmark test that highlights the momentum transfer through its effect on the granular flow deposit and on the amplitude of the generated wave. Moreover, the granular flow is gravitationally accelerated, which is a relevant aspect to test the behaviour of the numerical model.

## 2 Physical experiment of a granular landslide and wave

Miller et al. (2017) investigated the comprehensive phenomenon of landslide-generated tsunamis. In their study, the landslide, which consisted of a gravitationally accelerated granular flow, was simulated alongside the wave. The interaction between the two elements was of particular interest, and their reciprocal effects were highlighted. The momentum transfer obviously affected the wave behaviour but also influenced the landslide deposit.

Miller et al. (2017) and Mulligan et al. (2016) described the flume at Queen's University in Kingston (ON), Canada, where their physical experiments were conducted. This flume consisted of a 6.7 m aluminium plate inclined at an angle of 30°, followed by a 33-m-long horizontal section, and ending in a 2.4-m-long smooth impermeable slope of 27°; the width of the flume was 2.09 m. Nine different scenarios were tested, in which the water depth was varied from h = 0.05 to 0.5 m; one of these scenarios was tested without water. For each scenario, 0.34 m$^3$ (510 kg) of granular material was released from a box at the top of the slope. The granular material consisted of 3-mm-diameter ceramic beads, which had a material density of 2400 kg/m$^3$, a bulk density of 1500 kg/m$^3$, a static critical state friction angle of 33.7°, and a bed friction angle of 22°. The flat floor of the flume was composed of concrete. The bed friction angle was estimated to be approximately 35°. The wave amplitudes were determined by 5 probes, and the slide characteristics were captured with a high-speed camera (Cam 1).

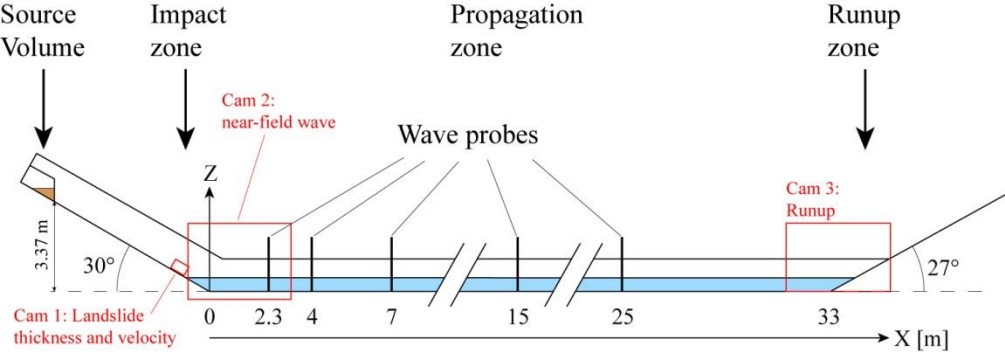

**Figure 1: Sketch of the flume used in the physical model and implemented in the numerical model (modified from Miller et al., 2017).**

## 3 Numerical model

The numerical model presented in this paper attempts to reproduce the experiment presented in Miller et al. (2017). The models for both the tsunami and the landslide simulations are based on shallow water equations. The two layers are



computed simultaneously (i.e., in the same iteration). The landslide layer is computed first and is considered as a bed change from the water layer (at each time step). The transfer of momentum also occurs at every time step.

## 3.1 Depth-averaged models

The model is based on two-dimensional shallow water equations:

$$U_t + F(U)_x + G(U)_y = S(U),$$   *eq. 1*

where $U$ is the solution vector and $F$ and $G$ are the flux vectors. These vectors are defined as follows:

$$U = \begin{bmatrix} H \\ Hu \\ Hv \end{bmatrix},$$   *eq. 2*

$$F = \begin{bmatrix} Hu \\ Hu^2 + \frac{1}{2}gH^2 \\ Huv \end{bmatrix}, G = \begin{bmatrix} Hv \\ Huv \\ Hv^2 + \frac{1}{2}gH^2 \end{bmatrix},$$   *eq. 3*

where $H$ is the depth; $u$ and $v$ are the components of the depth-averaged velocity vector along the x and y directions, respectively; and $g$ is the gravitational acceleration. The source term $S$ differs for the landslide (Eq. 8) and tsunami models (Eq. 19). Thus, the two formulations of the source term are specifically described in their respective sections (sect. 3.1.1 and 3.1.2). The conservative discrete form is expressed as follows:

$$U_{i,j}^{n+1} = U_{i,j}^n - \frac{\Delta t}{\Delta x}\left[F_{i+\frac{1}{2},j} - F_{i-\frac{1}{2},j}\right] - \frac{\Delta t}{\Delta y}\left[G_{i,j+\frac{1}{2}} - G_{i,j-\frac{1}{2}}\right] + \Delta t S_{i,j}^n,$$   *eq. 4*

where $F_{i+\frac{1}{2},j}$ is the intercell numerical flux corresponding to the intercell boundary at $x = x_{i+\frac{1}{2}}$ between cells $i$ and $i+1$ and $G_{i,j+\frac{1}{2}}$ is the intercell numerical flux corresponding to the intercell boundary at $y = y_{j+\frac{1}{2}}$ between cells $j$ and j+1. The Lax-Friedrichs (LF) scheme defines these terms as follows (Franz et al. (2013); Toro (2001)):

$$F_{i+\frac{1}{2},j}^{LF} = \frac{1}{2}\left(F_{i,j}^n + F_{i+1,j}^n\right) + \frac{1}{2}\frac{\Delta x}{\Delta t}\left(U_{i,j}^n - U_{i+1,j}^n\right),$$   *eq. 5*

$$G_{i,j+\frac{1}{2}}^{LF} = \frac{1}{2}\left(G_{i,j}^n + G_{i,j+1}^n\right) + \frac{1}{2}\frac{\Delta y}{\Delta t}\left(U_{i,j}^n - U_{i,j+1}^n\right),$$   *eq. 6*

This numerical scheme is chosen because of its non-oscillatory behaviour, its capacity to withstand rough beds, and its simplicity. See Franz et al. (2013) for more information concerning the choice of this numerical scheme.





### 3.1.1 Landslide model

The simulation of granular flow utilizes widely used rheological laws, among which the most commonly encountered are the Coulomb, Voellmy, and Bingham rheological laws (Iverson et al., 1997; Hungr and Evans, 1996; Longchamp et al. 2015;

Pudasaini and Hutter, 2007; Pudasaini, 2012; Kelfoun, 2011, McDougall, 2006; Pouliquen and Forterre, 2001). The continuum equations used are the previously described equations. The source term $S$ specifically governs the forces driving the landslide propagation:

$$S = \Delta t \begin{bmatrix} 0 \\ GR_x/\rho_s + P_x/\rho_s - T_x/\rho_s - M_{Tsx}/\rho_s \\ GR_y/\rho_s + P_y/\rho_s - T_y/\rho_s - M_{Tsy}/\rho_s \end{bmatrix}. \qquad \text{eq. 7}$$

where $\rho_s$ is the landslide bulk density, $T$ is the total retarding stress, and $M_{Ts}$ is the momentum transfer from the water to the

sliding mass. The driving components of gravity $GR$ and pressure term $P$ are defined as follows (Pudasaini and Hutter, 2007):

$$GR_x = \rho g H_s \sin \alpha_x \qquad GR_y = \rho g H_s \sin \alpha_y \qquad \text{eq. 8}$$

$$P_x = \rho g K_x \frac{\partial}{\partial x}\left(H_s{}^2 \cos \alpha_x\right) \quad P_y = \rho g K_y \frac{\partial}{\partial y}\left(H_s{}^2 \cos \alpha_y\right) \qquad \text{eq. 9}$$

where $H_s$ is the landslide thickness, $\alpha$ is the bed slope angle, and $K$ is the earth pressure coefficient. The density $\rho$ is the

relative density of the landslide. This means that $\rho$ is equal to the density of the slide $\rho_s$ when the slide is subaerial and $\rho = \rho_s\text{-}\rho_w$ (density of the water) when the slide is underwater (Kelfoun et al., 2010; Skvortsov, 2005). Since each term is divided by $\rho_s$ in Eq. (7), the relative nature of the density becomes effective. The total retarding stress $T$ ($T_x$, $T_y$) is composed of the resisting force (frictional resistance) between the landslide and the ground. $T$ differs depending on the chosen rheological law.

The simple Coulomb frictional law $Coul$ (MacDougall, 2006; Kelfoun et al., 2010; Kelfoun, 2011; Pudasaini and Hutter, 2007; Longchamp, 2015) is defined as follows:

$$T_x = Coul_x = \rho H_s(g \cos \alpha_x) \tan \varphi_{bed} \frac{u_s}{\|V\|} \qquad \text{eq. 10}$$

$$T_y = Coul_y = \rho H_s(g \cos \alpha_y) \tan \varphi_{bed} \frac{v_s}{\|V\|} \qquad \text{eq. 11}$$

where $\varphi_{bed}$ is the bed friction angle and $V$ ($u_s$, $v_s$) the slide velocity.

The Coulomb rheology can be used considering flow with either isotropic or anisotropic internal stresses. This difference is handled with the use of the earth pressure coefficient $K$ (Hungr and McDougall, 2009; Kelfoun, 2011, Iverson and





Denlinger, 2001; Pudasaini and Hutter, 2007). For the isotropic case, $K$ is set to 1 (Kelfoun, 2011). In the anisotropic case, $K$ designates whether the flow is in compression (positive sign) or in dilatation (negative sign), for which the coefficients are denoted as $K_{passive}$ or $K_{active}$ (Hungr and McDougall, 2009; Pudasaini and Hutter, 2007), respectively. These terms are defined
as follows:

$$K_{act/pass} = 2\frac{1\pm[1-(cos\,\varphi_{int})^2(1+(tan\,\varphi_{bed})^2)]^{\frac{1}{2}}}{(cos\,\varphi_{int})^2} - 1 \qquad\qquad eq.\ 12$$

$$K_x = \begin{cases} K_{active,} & \frac{\partial u_s}{\partial x} > 0 \\ K_{passive,} & \frac{\partial u_s}{\partial x} < 0 \end{cases} \quad K_y = \begin{cases} K_{active,} & \frac{\partial v_s}{\partial y} > 0 \\ K_{passive,} & \frac{\partial v_s}{\partial y} < 0 \end{cases} \qquad eq.\ 13$$

where the variable $\varphi_{int}$ is the internal friction angle.

The Voellmy rheology *Voel* combines Coulomb frictional rheology with a turbulent behaviour:


$$T_x = Voel_x = Coul_x + \rho\frac{u_s\|V\|}{\xi} \qquad\qquad eq.\ 14$$

The first term is the Mohr-Coulomb frictional law, whereas the second term, which was originally presented by Voellmy (1955) for snow avalanches, acts as drag and increases the resistance with the square of velocity. The turbulence coefficient $\xi$ corresponds to the square of the Chézy coefficient, which is related to the Manning coefficient $n$ by $C = H_s^{1/6}/n$ (MacDougall, 2006, p.76). The turbulence coefficient presented in Kelfoun (2011) is equivalent to the inverse of the
turbulence coefficient presented herein times g ($1/(\xi g)$). However, the physical basis of the Voellmy rheology is questionable (Fisher et al., 2012).

The Bingham rheology combines plastic and viscous rheological laws and is defined as follows (Skvortsov, 2005):

$$Plast_x = \frac{u_s}{\|V\|}T_0(1 + d_1) \qquad\qquad eq.\ 15$$

$$Visc_x = \frac{2\mu_s u_s}{H_s(1-(d1/3))} \qquad\qquad eq.\ 16$$


$$T_x = Bingham_x = Plast_x + Visc_x \qquad\qquad eq.\ 17$$

where $T_0$ is the yield stress, which is the stress to overcome for the slide to start or under which the slide stops; $\mu$ is the dynamic viscosity; and $d_1$ is the relative thickness of the shear layer. The latter variable is used to mimic the behaviour of Bingham flow that contains two distinct layers: a solid layer (the plug zone) and a shear layer (the shear zone, $d_1$). The determination of $d1$ can be performed automatically (e.g., Skvortsov, 2005), but in this study, the use of a constant value
provided better results.





### 3.1.2 Tsunami model

For the tsunami model, the source term $S$ includes a consistency term and a momentum transfer term and is defined as follows:

$$S = \Delta t \begin{bmatrix} 0 \\ -H_w g 0.5 \frac{\partial}{\partial x}(B) + \frac{M_{Twx}}{\rho_w} \\ -H_w g 0.5 \frac{\partial}{\partial y}(B) + \frac{M_{Twy}}{\rho_w} \end{bmatrix}. \qquad \text{eq. 18}$$

where $H_w$ is the water thickness, $B$ *is* the bed elevation (including the thickness of the sliding mass), and $M_{Tw}$ is the momentum transfer from the slide to the water. The first term confers consistency to the model, which has been validated in Franz et al. (2013). The second term is the momentum transfer between the landslide and the water.

The wet-dry transition is realized by an ultrathin layer of water $h_{min}$ that covers the whole topography (or the dry state). This permits the avoidance of zeros in the water depth array. Nevertheless, such a situation would lead to water flowing down the

slopes after some iteration. Thus, to prevent this numerical artefact, the thin layer is governed by viscous equations (Turcotte and Schubert, 2002):

$$Q_x = \begin{bmatrix} F_i^{LF} \\ \rho_w \frac{q_x^2}{H_w} + \rho g \frac{1}{2} H_w \\ \frac{\rho_w u_w v_w}{H_w} \end{bmatrix} \qquad \text{eq. 19}$$

where

$$q_x = -\left(\sin \alpha_x + \frac{\partial}{\partial x}(H_w)\right)\frac{\rho_w g H_w^3}{3\mu_w} \qquad \text{eq. 20}$$

A threshold $Re_{tr}$ is set for a Reynolds number that determines which set of equations (between viscous equations and shallow water equations) is used at each location in the tsunami model:

$$Re = \frac{\rho_w u_w H_w}{\mu_w} \qquad \text{eq. 21}$$

$$F_{i+\frac{1}{2}}^{LF} = \begin{cases} F_{i+\frac{1}{2}}^{LF}, & Re > Re_{tr}, \quad H_w > h_{min} \\ Q_x, & Re < Re_{tr}, \quad H_w < h_{min} \end{cases} \qquad \text{eq. 22}$$



### 3.1.3 Momentum transfer

The interaction between the landslide and the water proposed by Kelfoun et al. (2010) is based on a formulation of drag both normal and parallel to the displacement. This formulation involves two coefficients that need to be set manually, which is a manipulation this study aims to avoid. Moreover, they claim that their equation is a rewritten form of the equation presented in Tinti et al. (2006). However, in the latter, the depth of the landslide $Hs$ is taken into account, whereas in Kelfoun et al. (2010), they account for the gradient of the landside depth.

In Xiao et al. (2015), the so-called "drag-along" approach also entails undesired (from our point of view) user-defined coefficients and, when tried in our code, never fit the experiment data.

Regarding those two unsatisfying approaches, we decided to try an unconventional method. Based on a semi-empirical approach that fits the experimental data, the implementation of momentum transfer in our code is inspired by the simple perfectly elastic collision principle. This principle does not generally apply in fluid dynamics, but it is relevant because 1)

the kinetic energy of the system is conserved, 2) a true interaction between particles is no longer possible in a model based on shallow water equations (free surface – no third dimension), and 3) the exchange of momentum between the landslide mass and the water is perfectly symmetric.

As a reference case, we consider velocity changes during the elastic collision of two rigid bodies. The conservation of momentum in elastic collision is given by the following expression:

$$m_s u_{sb} + m_w u_{wb} = m_s u_{sa} + m_w u_{wa} \qquad\qquad eq.\ 23$$

As the kinetic energy is also conserved, the following constraint applies:

$$\tfrac{1}{2} m_s u_{sb}^2 + \tfrac{1}{2} m_w u_{wb}^2 = \tfrac{1}{2} m_s u_{sa}^2 + \tfrac{1}{2} m_w u_{wa}^2 \qquad\qquad eq.\ 24$$

where $u_w$ and $u_s$ are the velocities for the 'water' and the 'slide' masses, respectively (subscript $b$ = before collision and subscript $a$ = after).

We assumed that the mass before collision remained constant after collision. Under this simplifying assumption, the two conservation equations can be used to solve for the two velocities after collision:

$$u_{sa} = \frac{(m_s u_{sb} - m_w u_{sb} + 2 m_w u_{wb})}{(m_s + m_w)} \qquad\qquad eq.\ 25$$

$$u_{wa} = \frac{(2 m_s u_{sb} - m_s u_{wb} + m_w u_{wb})}{(m_s + m_w)} \qquad\qquad eq.\ 26$$

The discrete "finite control volume" masses $m$, having lateral lengths $dx$ and $dy$, are defined as follows:

$$m_s = \rho_s * dx * dy * H_s \qquad\qquad eq.\ 27$$




$$m_w = \rho_w * dx * dy * H_w \qquad\qquad eq.\ 28$$

Using this notation, the above expressions for velocity changes during collision can be rearranged in a form similar to the time- and space-discretized depth-averaged momentum equation:

$$\frac{\partial(\rho_s H_s u_s)}{\partial t} \approx \rho_s H_s (u_{sa} - u_{sb})/dt \ = \ \frac{2}{\left(\frac{1}{H_w \rho_w} + \frac{1}{H_s \rho_s}\right)}(u_{sb} - u_{wb})/dt \qquad eq.\ 29$$


$$\frac{\partial(\rho_w H_w u_w)}{\partial t} \approx \rho_w H_w (u_{wa} - u_{wb})/dt = \frac{2}{\left(\frac{1}{H_w \rho_w} + \frac{1}{H_s \rho_s}\right)}(u_{sb} - u_{wb})/dt \ eq.\ 30$$

where $dt$ is the time discretization. Note that $dx$ and $dy$ are cancelled out. The right-hand sides of Eqs. (29 & 30) represent the momentum exchange source terms during collision.

The momentum transfer during the rigid collision reference case was modified by a 'shape factor' $S_F$ as a fitting parameter to
reproduce the laboratory experiments from Miller et al. (2017), resulting in the following expressions:

$$M_{Ts} dt = S_F \frac{2}{\left(\frac{1}{H_w \rho_w} + \frac{1}{H_s \rho_s}\right)}(u_{wb} - u_{sb}) \qquad\qquad eq.\ 31$$

$$M_{Tw} dt = - S_F \frac{2}{\left(\frac{1}{H_w \rho_w} + \frac{1}{H_s \rho_s}\right)}(u_{sb} - u_{wb}) \qquad\qquad eq.\ 32$$

The shape factor $S_F$ is defined from experiments to match the wave amplitude and the landslide deposit. This shape factor consists of a non-dimensional value that depends on the ratio between the maximum landslide thickness $s_{max}$ at impact and
the still water level $h$:

$$SF = 0.145(s_{max}/h)^{1.465} \qquad\qquad eq.\ 33$$

The choice of the values presented in Eq. (33) is a compromise to accurately fit the wave amplitude and the landslide deposit considering different still water levels. Some values investigated in this process are presented in Fig. 2.

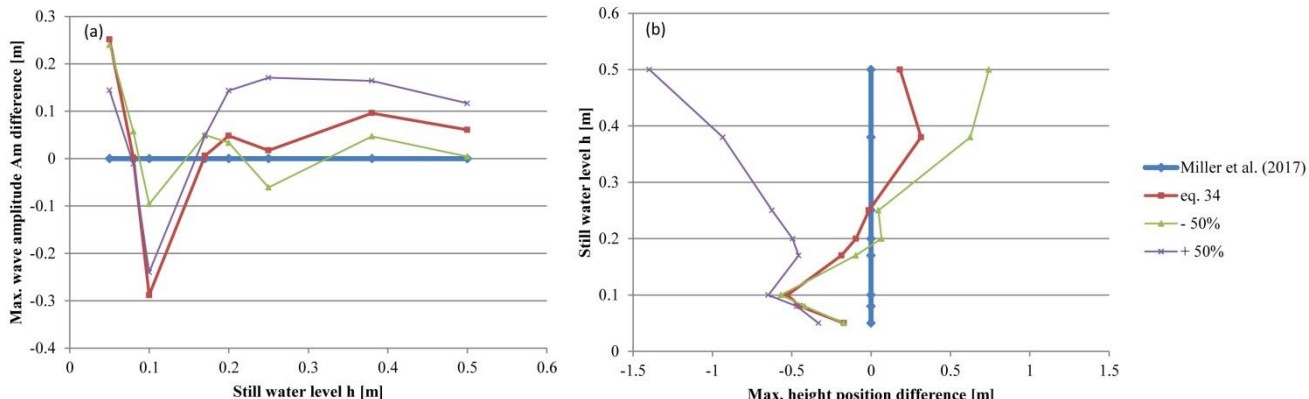

Figure 2: (a) Graph showing the difference between the $A_m$ values from Miller et al. (2017) (blue line) and the $A_m$ values from the numerical simulation using different values in Eq. (33). Note that the value used in Eq. (33) (red line) is not the best fitting curve. (b) Graph showing the difference between the positions of the apex of the landslide deposits observed in Miller et al. (2017) (blue line) and the positions obtained from the numerical simulations. The best fitting values used in Eq. (33) are the values presented. The final choice of values is a compromise to accurately fit the wave amplitude and the landslide deposit.

The set of equations presented in Mulligan and Take (2016) also describes the process of momentum transfer. However, the differences between their approach and our approach make comparisons difficult. On the one hand, our method is performed through time discretization, whereas the approach adopted by Mulligan and Take (2016) is performed in one "time step". Although this difference does not make the two approaches inherently incomparable, the equation proposed by Mulligan and Take (2016) defines the near-field maximum wave amplitude $A_m$ as a function of the slide, the apparatus and the water body parameters (such as $\rho_s$, $\alpha$, $s$, $v_s$, and $h$) whereas our equations define momentum transfer without change in height (Eqs. 25 & 26). In our code, the height change is obtained after, while solving the depth-averaged equations.

## 4 Results

The landslide and the tsunami models are computed in 2D ($x$ and $y$), whereas the results, such as the landslide thickness or the water elevation, are represented visually in the third dimension ($z$). In this study, everything is computed as presented in Fig. 3, but the interpretations of the results are done through longitudinal cross-sections across the centre of the numerical flume (Fig. 4).




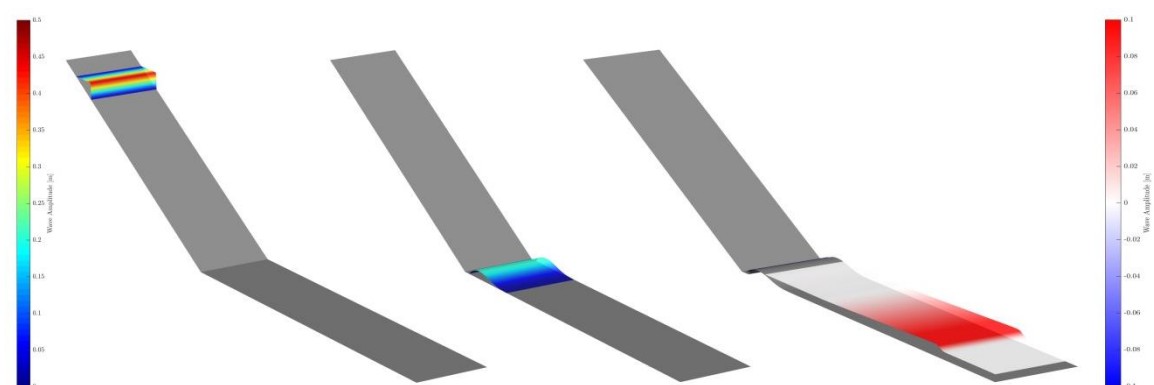

**Figure 3: 2.5D numerical representation of the near-field section of the flume with water depth h = 0.2: (left) initial condition of the landslide (water level not displayed), (centre) landslide deposit (water level not displayed), and (right) landslide deposit (not coloured) with the generated wave.**

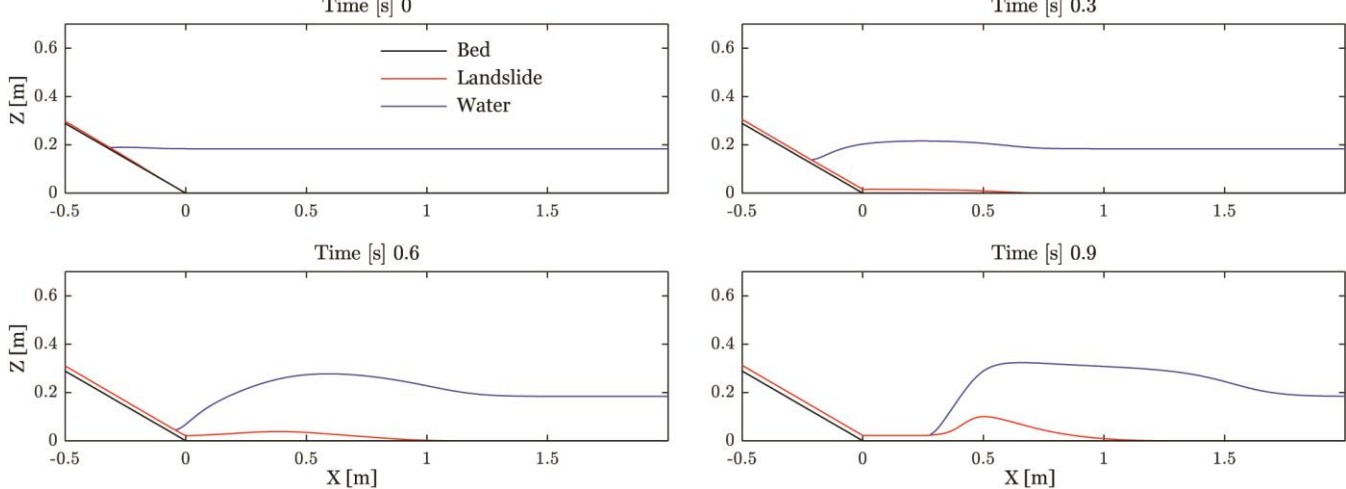

**Figure 4: Profile view of the landslide and water surface during the generation of the wave.**

## 4.1 Landslide

This section presents the results concerning the granular landslide. Furthermore, this section discusses the behaviour of the landslide propagation using different rheological laws and the effect of the water depth on the landslide deposit.

### 4.1.1 Dry case

The dry case investigates the propagation of the granular material using various rheological laws. The rheological laws implemented herein are the Voellmy, Coulomb (flow with isotropic and anisotropic internal stresses) and Bingham





rheological models. The velocities, the thicknesses and the deposit shapes obtained through the numerical simulation are

compared to those data obtained from the physical experiment to identify and select the best solution.

**Table 1: Rheological parameters used for the different rheological laws.**

| | Shear zone rel. thick. [-] | Yield stress [Pa] | Dynamic viscosity [Pa·s] | Turbulence coeff. [m/s²] | Bed friction angle alu. [°] | Bed friction angle concr. [°] | Int. friction angle [°] |
|---|---|---|---|---|---|---|---|
| | $d_1$ | $T_0$ | $\mu$ | $\xi$ | $\Phi_{bed}$ | $\varphi_{bed}$ | $\varphi_{int}$ |
| Bingham | 0.6 | 12 | 1.6 | - | - | - | - |
| Voellmy | - | - | - | 250 | 11 | 35 | - |
| Coulomb (iso.) | - | - | - | - | 22 | 35 | - |
| Coulomb (aniso.) | - | - | - | - | 22 | 35 | 33.7 |

The Bingham rheology is set by best fit with a shear zone relative thickness $d_1$ of 0.6, a yield stress $T_0$ of 12 Pa, and a

dynamic viscosity $\mu$ of 1.6 Pa·s.

Concerning the Voellmy rheology, the determination of the turbulence coefficient $\zeta$ is performed by trial and error to obtain

the best fit (back analysis). Thus, the turbulence coefficient $\zeta$, as described in Hungr and Evans (1996) and in McDougall

(2006), is set to 250 m/s². The bed friction angle $\varphi_{bed}$ of 22°, given in Miller et al. (2017), was reduced to 11°. Indeed, the

Voellmy rheology uses significantly lower values (Hungr and Evans, 1996). This study uses the same ratio (~0.5) between

"classical" $\varphi_{bed}$ and "Voellmy" $\varphi_{bed}$ as the one presented in Hungr and Evans (1996) for cases with similar friction angles

and turbulence coefficients ("Voellmy" $\varphi_{bed}$ = 11°).

Regarding the two Coulomb models, we use the bed friction angle $\varphi_{bed}$ of 22°, as measured in the physical experiment

(Miller et al, 2017). In addition, the anisotropic Coulomb rheological model considers the internal friction angle $\varphi_{int}$, which

is 33.7° (Miller et al, 2017).

In Miller et al. (2017), the velocity and the thickness of the landslide at impact are estimated through high-speed camera

footage analysis with a still water depth $h$ of 0.25 m. To measure the same variables of the simulated granular flow, the

values are recorded using a corresponding window (Cam 1, Fig. 1).

Figure 5 shows the temporal evolution of the flow thickness and velocity captured at the numerical equivalent location of

Cam 1 (Fig. 1). The numerical models do not fit the physical simulation very well. This poor fit can be explained by the

diffuse nature of a granular flow boundary that is not replicable by the shallow water assumption (continuum mechanics) and

by the absence of expansion in the numerical moving mass. Nevertheless, the results from the numerical and physical models

are on the same order of magnitude, which permits globally validating the different numerical models but does not allow

discrimination between them.





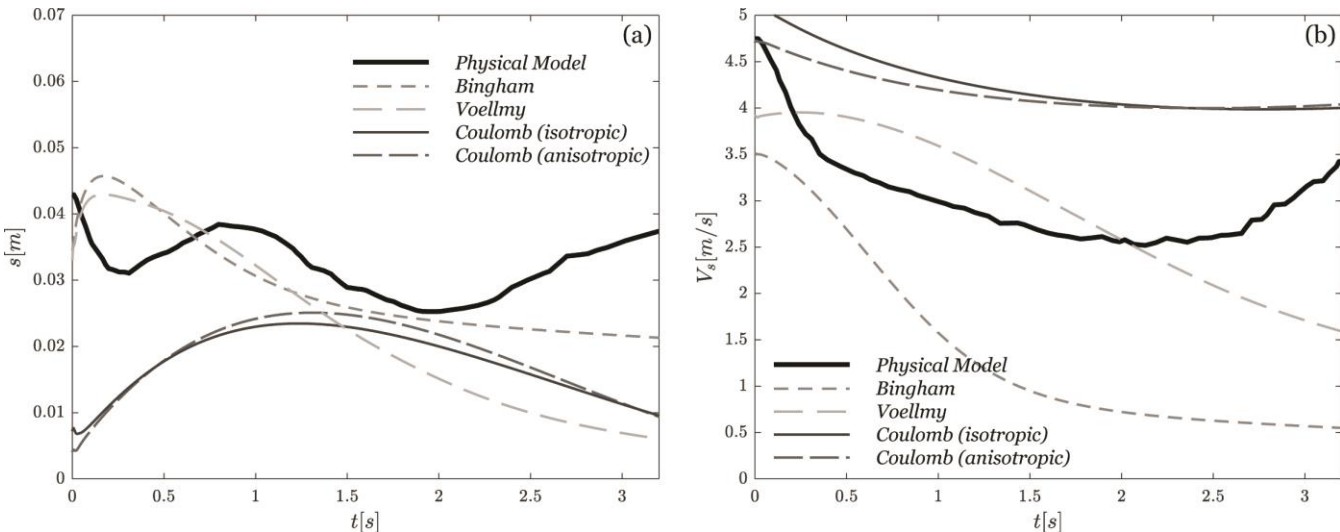

**Figure 5: Landslide properties at impact (numerical equivalent location of Cam 1) with a still water depth h of 0.25 m. (a) Time series of slide thickness for different rheological laws (numerical model) compared with the mean thickness in the physical model. (b) Time series of depth-averaged slide velocity for different rheological laws (numerical model) vs. the mean velocities in the physical model (modified from Miller et al., 2017).**

Consequently, analysing the deposit (Fig. 6) is the way to identify the best fitting rheological model. The Bingham rheological model does not correctly reproduce the shape of a granular deposit. The Voellmy model performs better than the Bingham model in this respect, but in comparison with the two Coulomb frictional models, the Voellmy model is not satisfactory. Indeed, the two Coulomb rheological models (anisotropic and isotropic) fit the best with the observed deposit, which was expected because frictional rheological laws are typically developed to describe granular flows. The rear parts of the deposits are correctly located, whereas the fronts are slightly too distant. However, this imperfection is negligible and could be attributed to numerical diffusion. The deposit simulated with the isotropic Coulomb model is slightly closer to the real deposit than that simulated with the anisotropic model; this method has the advantage of being simple (only the bed friction angle $\varphi_{bed}$ is implemented).





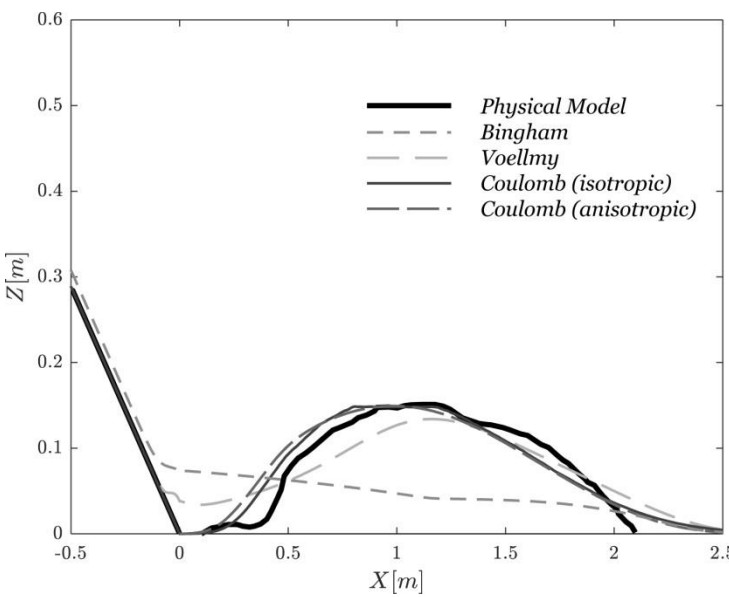

**Figure 6: Cross-section of the landslide deposit using different rheological laws compared with the deposit in the physical experiment.**

Since the velocities and the thickness are of a realistic magnitude, the deposit shapes of the two Coulomb models correctly reproduce the real case. Furthermore, due to the ease of implementation, the isotropic Coulomb model is the rheological model of choice. In the sect. 4.1.2, this model is used to study the wet cases.

### 4.1.2 Wet cases

This section investigates the interaction between the landslide and the water. More precisely, this section investigates the
effect of the momentum transfer on the deposit shape for different water levels. Figure 7a shows the results for still water depths $h$ of 0.05, 0.08, and 0.1 m. The deposits resulting from the numerical simulation (solid lines) are compared with the physical model observations (dashed lines), which shows a rather good similarity when focusing on the height of the piles. The numerical deposition shape for a still water depth of 0.05 m fits well the physical shape, also regarding the spread. Concerning still water depths $h$ of 0.08 and the 0.1 m, the numerical granular flows stop more distantly than the real flows.
At still water depths of 0.17, 0.2, and 0.25 m (Fig. 7b), the numerical and physical results are in good agreement; however, the "tails" and the apexes of the deposits are located slightly farther away in the numerical simulation.




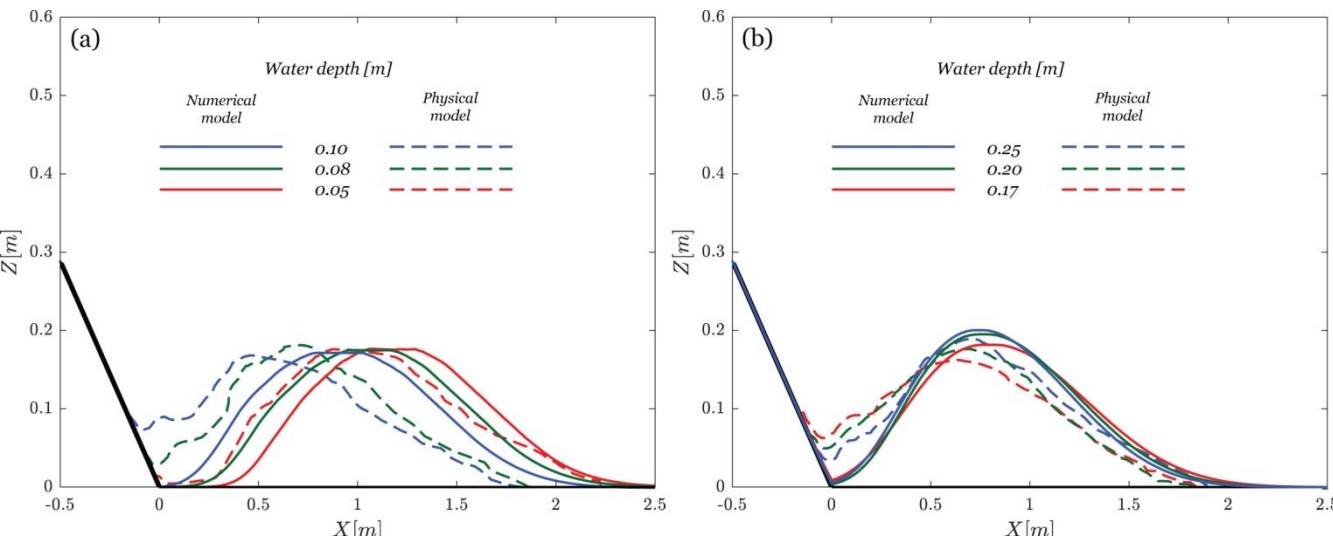

**Figure 7: Cross-section of the granular flow deposit for different still water depths h (0.05 to 0.25 m). The dashed lines represent the physical experiment observations (modified from Miller et al., 2017), whereas the solid lines represent the results of the numerical model.**

In contrast, for still water depths $h$ of 0.38 and 0.5 m (Fig. 8), the deposit shapes obtained by the numerical simulations stop slightly ahead of the real deposits. Nevertheless, the deposits are of equivalent heights.

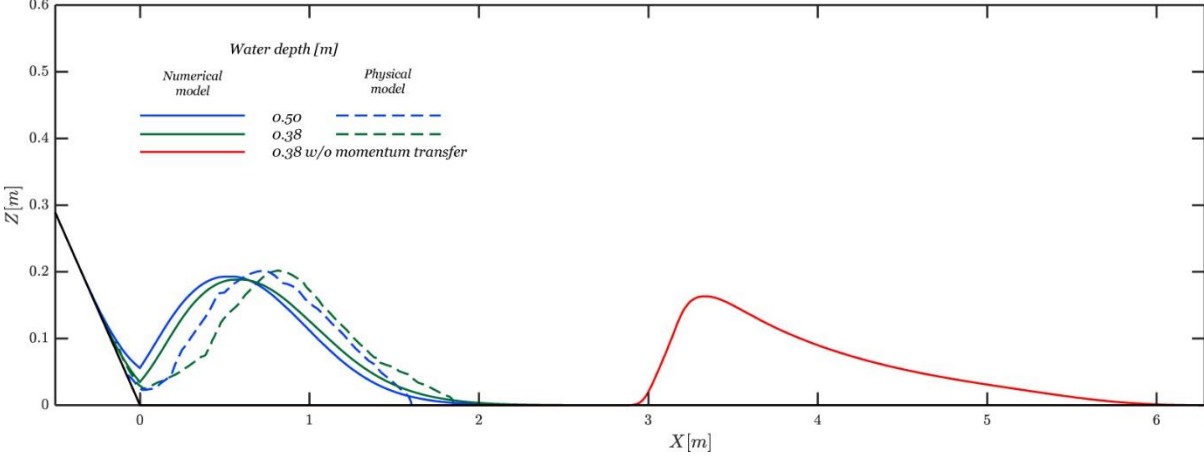

**Figure 8: Cross-section of the granular flow deposit for two still water depths h (0.38 and 0.5 m). For comparison, the red line illustrates the landslide deposit for a still water depth h of 0.38 m without momentum transfer. The dashed lines represent the physical experiment observations, whereas the solid lines represent the results of the numerical model (modified from Miller et al., 2017).**

The momentum transfer acts correctly on the granular flow as the global correspondence between the numerical and physical deposition pattern is good. In fact, it is the combination of the momentum transfer (Eqs. 31 & 32) with the relative density $\rho$ (Eqs. 7 through 11) that performs well. This is highlighted by the simulated granular landslide without momentum transfer, which travels excessively far (Figure 8 8, red line). The travel distance in this case is even greater than that in the dry case



(result of the isotropic Coulomb model presented in Fig. 6) due to the effect of the relative density $\rho$. Indeed, the "drop in density" when the granular flow enters the water body reduces the total retarding stress $T$ in particular (Eqs. 10 & 11; alongside $P$ (Eq. 9) and $GR$ (Eq. 8), which is negligible on flat surfaces). It is worth noting that without momentum transfer

or relative density, the model would lead to the same deposit as the dry case.

## 4.2 Wave

This section investigates the second aspect of the momentum transfer between the slide and the water: its effect on the generated wave. This effect is analysed for different still water depths $h$ (0.05, 0.1, 0.2, and 0.5 m) through probes located at different distances from the bottom of the slope (2.3, 15, and 23 m; Fig. 9). Concerning the case with the smallest still water

depth ($h = 0.05$ m), the numerical simulation reproduces the wave observed in the physical experiment very well in terms of amplitude and timing at each probe. Note that the simulated wave is taller than the real wave in the very near field (2.3 m gauge). For a still water depth $h$ of 0.1 m, the timing is good at the 2.3 m gauge, but, as previously described, the numerical wave is taller. Concerning the gauges at 15 and 23 m, the wave celerity is faster and its amplitude is smaller in the numerical simulation than in the physical experiment. Moreover, the wave train observed in the physical model is non-existent in the

numerical model. Except for the equivalence in amplitude in the near field, the same observations apply for a still water depth $h$ of 0.2 m. For a still water depth of 0.2 m, a reflected wave is present at the 23 m gauge after approximately 23 s. The numerically simulated wave arrives slightly earlier than the observed wave. Concerning the case of a still water depth $h$ of 0.5 m, the simulated wave is slightly smaller than the real wave, and the reflected wave (at 28, 23, and 18 s) is visible at the 3 gauge locations with a good correspondence in terms of time.




**Figure 9: Time series of the relative water surface elevation η/h for different still water depths _h_ (.0.05, 0.10, 0.20 and 0.50 m) observed at different wave probes/gauges (2.3, 15, and 25 m). The dashed lines represent the physical experiment observations, whereas the solid lines represent the results of the numerical model (modified from Miller et al., 2017).**

### 4.2.1 Runup

Figure 10 presents the comparison of the runup height $R$ as a function of the maximum amplitude $A_m$ at the 25 m gauge between the physical experiments and the numerical simulation. Even if the wave is breaking with water depths of 0.38 and 0.5 m in the physical experiment, the match is adequate between the physical experiment and the simulation. Moreover, the underestimation of $A_m$ for the 0.1-0.25 m water level can be explained by the wave train that occurs in the flume but not in the numerical simulation. We can underline that there is a better match with the runup height than with the wave amplitude.


This phenomenon can be explained by the fact that when a wave train is present, it produces a higher frontal wave but that

the volume of displaced water is similar (trough and crest compensate) and hence a similar runup height.

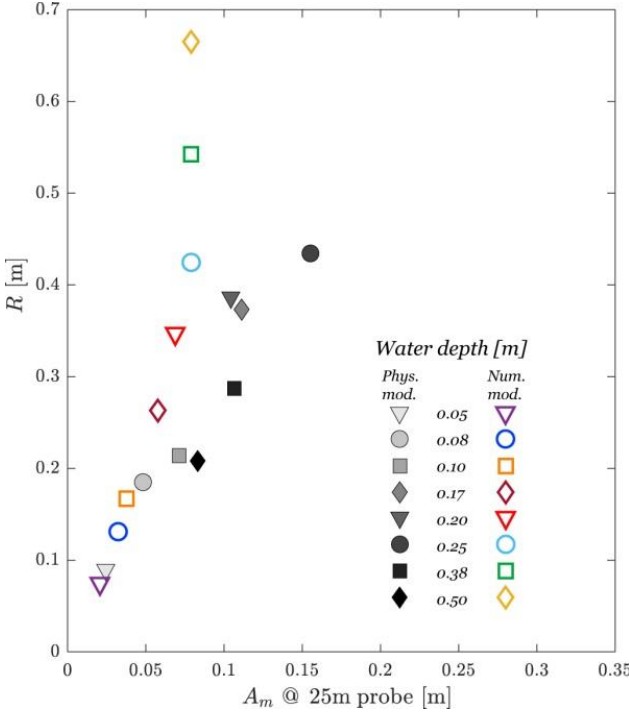

**Figure 10: Runup on a smooth impermeable slope of 27°. Runup height as a function of the incident wave maximum amplitude A$_m$ at the 25 m gauges. The coloured hollow shapes represent the results of the numerical model, whereas the solid greyscale shapes**
**are the observed values from the physical model (modified from Miller et al., 2017).**

**4.2.2 Impulse product parameter**

Heller and Hager (2010) proposed a relationship between the landslide characteristics and the near-field maximum amplitude

of the generated wave through the concept of the impulse product parameter $P$. The impulse produce parameter includes the

governing parameters related to the landslide and the still water depth. The maximum wave amplitude can be predicted as a

function of $P$ through Eq. (37). This approach is relevant to our study because it inherently considers the momentum transfer

occurring during wave generation. The following values are captured at the impact zone (Cam 1, Fig. 1) for the sliding mass

and in the near-field area for the wave. The relative maximum near-field wave amplitude $A_m$ defined by the following

expression:

$$A_m = 0.25\, Fr^{1.4}\, S^{0.8} \qquad\qquad eq.\ 34$$

where $S$ is the relative landslide thickness and $Fr$ is the Froude number, which are defined as follows:

$$S = s_{max}/h \qquad\qquad eq.\ 35$$




$$Fr = u_s/\sqrt{(gh)} \qquad \qquad eq.\ 36$$

The relationships between the impulse product parameter and the Froude number and the relative landslide thickness were found empirically through a large set of tests based on different reservoir and landslide setups (Fritz et al., 2004). The

impulse product parameter $P$ defined by Heller and Hager (2010) is expressed as follows:

$$P = Fr\ S^{1/2}\ M^{1/4}\ \left\{cos\left[(6/7)\alpha\right]\right\}^{1/2} \qquad \qquad eq.\ 37$$

where $M$ is the relative landslide mass, which is defined by the following expression:

$$M = m_s/(\rho_w bh^2) \qquad \qquad eq.\ 38$$

where $m_s$ is the landslide mass and $b$ is the flume width. The near-field relationship between the $P$ and $A_m$ is defined as

follows (Heller and Hager, 2010):

$$A_M = \frac{4}{9}P^{4/5} \qquad \qquad eq.\ 39$$

Figure 11 shows this relationship with the results of Miller et al. (2017) alongside the results of the present study. The two dashed lines represent the same relationship ± 30%. A large set of data collected in flume experiments (Fritz et al., 2004; Heller and Hager, 2010) falls between those limits for $P<9$.

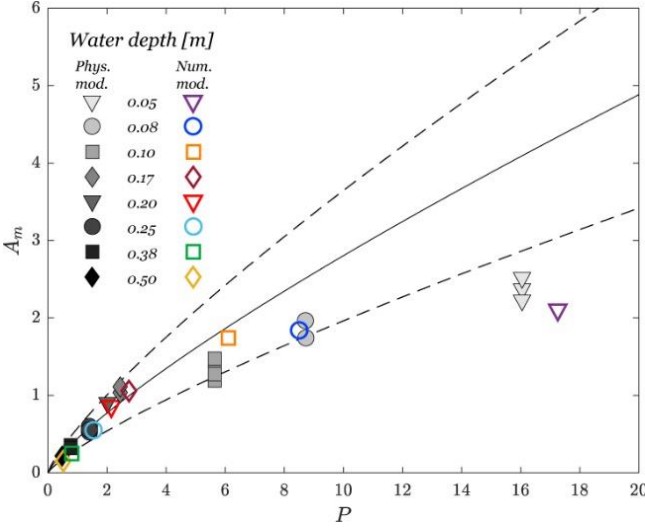


**Figure 11: Maximum relative wave amplitude $A_m$ as a function of the impulse product parameter P. (solid line) $A_M$ from Eq. (39), (dashed lines) $A_M$ from Eq. (39) ± 30%, (solid shapes) data from the physical model (Miller et al. 2017), and (hollow shapes) data from the numerical simulation (this study).**





The near-field relationships between $P$ and $A_m$ obtained in the present study correspond very well with those obtained by
Miller et al. (2017) (Fig. 11). On the other hand, for $P<9$ (as originally presented in Heller and Hager, 2010), the results of
this study are located within a range of ±30%.

### 4.3 Discussion

The numerical model displays a taller wave in the near field, which can be explained by the fact that the model does not
reproduce the breaking of the wave. This discrepancy is inherent to the shallow water model and its two-dimensional nature.
This finding is supported by the fact that this phenomenon is observed for still water depths $h$ of 0.05-0.17 m, which are
depths prone to wave breaking in the Miller et al. (2017) experiments.

The physical experiments produce wave trains for water levels of 0.1 and 0.2 m. This phenomenon is not reproduced by the
numerical model; this finding could be explained by the absence of breaking in the unstable numerical waves, which is the
cause of the aforementioned train (Miller et al. 2017). It is clear that the complex interaction between the landslide and the
presence of possible small-scale backwash during generation is not the cause. Indeed, the "unique" wave at the 2.3 m probe,
which is farther than the slide deposit, verifies that the apparition of the train occurs after the wave generation (Miller et al.
2017). On the other hand, the front of the wave is very different. The "excess" volume of water at the front of the numerical
wave is also partially explained by the lack of energy dissipation that would occur during breaking. On average, the
simulated water level located at the wave train "match" the trough and the crests. For those cases, the imperfect
reproductions are, however, sufficiently close in terms of celerity and volume to be considered relevant. This consideration
was further confirmed by the good match of the reflected wave (23 s) and the measured runup (Fig. 10).

The general observation of the evolution of the wave (Fig. 9) shows that the decay occurring in the physical experiment is
present in the numerical simulation. This fact also strengthens the general validity of the whole numerical model.

Inherently, as the impulse product parameter values obtained through a wide set of experiments (Heller and Hager, 2010;
Miller et al., 2017) fall into an envelope of ±30%, our near-field results, which also fall into these limits, strongly confirm
the validity of our model and our momentum transfer approach.

### 5 Conclusions

The dry case shows that the two Coulomb rheological models (flow with isotropic or anisotropic internal stresses) correctly
reproduce the deposit observed in the physical model studied by Miller et al. (2017). The isotropic Coulomb model is the
simplest and easiest to implement and is chosen to study the wet case.

The numerical simulation of the wet case investigates the abilities of the model to correctly handle momentum transfer. This
case focuses on both the effect of the water on the landslide deposit and the effect of the landslide on the resulting wave.
These effects are investigated through different water levels, and it appears that the landslide deposit obtained by the
numerical simulations fits well with the physical model observations. On the other hand, the numerical waves behave



similarly to the waves in the physical model. Despite imperfections, the combined results from investigating these two effects permits us to consider that, overall, the model effectively handles the complex phenomenon occurring during the interaction between the landslide and the water. In addition, the choice to transfer the momentum through the simple perfectly elastic collision principle is verified to be relevant.

A comparison involving impulse product parameters particularly highlights that our model satisfactorily reproduces the
physical experiment of Miller et al. (2017). The values of $A_m$ versus $P$ presented in Heller and Hager (2010) are based on 223 sets of flume experiments performed by Fritz et al. (2002) and Zweifel (2004). Hence, the validity of our model is further strengthened by the fact that the results of our model also fit well with those experiments.

Finally, our model is validated by a benchmark test performed herein, as this approach is very simple to implement and is very efficient in terms of computational resources. Therefore, we consider our model as a tool of choice for the assessment
of landslide-generated tsunami hazards.

*Author contributions.* MF and MJ conceived the project. MF MJ RM and AT designed the benchmark tests. MF and YP developed the model code, RM ant AT provided the data and MF carried out the simulations. MF prepared the manuscript with review and editing from the all co-authors.


*Competing interests.* The authors declare that they have no conflict of interest.

*Acknowledgements.* The authors are thankful to Dr. Shiva P. Pudasaini for sharing his valuable insights about landslides impacting water bodies, particularly on the associated complexity of momentum transfer.

**Notation**

The following symbols are used in this paper:

| | | |
|---|---|---|
| $A_m$ | = | maximum amplitude of measured wave [m] |
| $A_M$ | = | theoretical maximum amplitude of near-field wave [m] |
| $B$ | = | bed elevation [m] |
| $b$ | = | flume width [m] |
| $C$ | = | Chézy coefficient [-] |
| $d_1$ | = | shear layer relative thickness [-] |
| $F$ | = | flux vector in x direction [-] |
| $Fr$ | = | Froude number [-] |
| $G$ | = | flux vector in y direction [-] |
| $g$ | = | gravitational acceleration [m/s$^2$] |





| | | |
|---|---|---|
| $GR$ | = | driving component of gravity [Pa] |
| $H$ | = | layer depth [m] |
| $h$ | = | still water level [m] |
| $h_{min}$ | = | minimum water thickness (ultrathin layer) [m] |
| $H_s$ | = | landslide thickness [m] |
| $H_{sa}$ | = | landslide thickness after collision [m] |
| $H_{sb}$ | = | landslide thickness before collision [m] |
| $H_w$ | = | depth of the water [m] |
| $H_{wa}$ | = | depth of the water after collision [m] |
| $H_{wb}$ | = | depth of the water before collision [m] |
| $K$ | = | earth pressure coefficient [-] |
| $LF$ | = | Lax-Friedrichs scheme |
| $M$ | = | relative landslide mass [-] |
| $m_s$ | = | landslide mass [kg] |
| $MT_s$ | = | momentum transfer (water->slide) [Pa] |
| $MT_w$ | = | momentum transfer (slide->water) [Pa] |
| $n$ | = | Manning roughness coefficient [-] |
| $P$ | = | pressure term [Pa] |
| $P$ | = | impulse product parameter [-] |
| $R$ | = | runup height [m] |
| $Re$ | = | Reynolds number [-] |
| $Re_{tr}$ | = | Re threshold [-] |
| $S$ | = | source term [-] |
| $S$ | = | relative landslide thickness [-] |
| $SF$ | = | shape factor for momentum transfer [-] |
| $s_{max}$ | = | maximum landslide thickness [m] |
| $T$ | = | total retarding stress [Pa] |
| $T_0$ | = | yield stress [Pa] |
| $U$ | = | Solution vector [-] |
| $u$ | = | velocity vector component in x direction [m/s] |
| $u_s$ | = | landslide velocity in x direction [m/s] |
| $u_{sa}$ | = | landslide velocity in x direction after collision [m/s] |
| $u_{sb}$ | = | landslide velocity in x direction before collision [m/s] |
| $u_w$ | = | water velocity in x direction [m/s] |
| $u_{wa}$ | = | water velocity in x direction after collision [m/s] |
| $u_{wb}$ | = | water velocity in x direction before collision [m/s] |





| $v$ | = | velocity vector component in y direction [m/s] |
| $V$ | = | full velocity vector [m/s] |
| $v_s$ | = | landslide velocity in y direction [m/s] |
| $v_w$ | = | water velocity in y direction [m/s] |
| $x$ | = | longitudinal coordinate [m] |
| $y$ | = | transverse coordinate [m] |
| $z$ | = | vertical coordinate [m] |
| $\alpha$ | = | bed slope angle [°] |
| $\Delta t$ | = | time step [s] |
| $\eta$ | = | wave amplitude [m] |
| $\mu_s$ | = | landslide dynamic viscosity [Pa s] |
| $\mu_w$ | = | water dynamic viscosity [Pa s] |
| $\xi$ | = | turbulence coefficient [m/s$^2$] |
| $\rho$ | = | relative density [-] |
| $\rho_s$ | = | landslide bulk density [kg/m$^3$] |
| $\rho_{sa}$ | = | landslide bulk density after collision [kg/m$^3$] |
| $\rho_{sb}$ | = | landslide bulk density before collision [kg/m$^3$] |
| $\rho_w$ | = | water density [kg/m$^3$] |
| $\rho_{wa}$ | = | water density after collision [kg/m$^3$] |
| $\rho_{wb}$ | = | water density before collision [kg/m$^3$] |
| $\varphi_{bed}$ | = | bed friction angle [°] |
| $\varphi_{int}$ | = | internal friction angle [°] |

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
