# Peer review of "An efficient two-layer landslide-tsunami numerical model: effects of momentum transfer validated with physical experiments of waves generated by granular landslides"

_Natural Hazards and Earth System Sciences, 2019_

## Referee Comment (RC1) · Anonymous Referee #1 · 17 Feb 2020

Comments for the Authors:

The present paper deals with the numerical modelling of landslide-generated tsunamis by using a two-layer model, based on the shallow water equations, and a novel (semi-empirical) momentum transfer approach based on the perfectly elastic collision principle. The topic addressed by the present paper is a relevant one in the landslide-generated tsunamis research field. The scientific quality of the paper is good. The paper is in general well written and structured, defining clearly the objectives and describing the methodology and the results. Nevertheless, few points need to be clarified/discussed with more details. This Reviewer believes that the paper can be of interest for the Scientific Community after a minor review process, in particular more details and discussion should be provided in presenting methods and results (see major and minor points).

Please find enclosed a detailed list of major and minor issues.

Major Points:

-Point-0, L37-43 A more detailed discussion of more complex and time consuming numerical models (based on the RANS equations, e.g. Abadie et al., 2010; Clous & Abadie, 2019) is needed. This would help the Readers in comparing the approach proposed by the Authors with the ones available in the scientific literature (see also Point-1).

-Point-1, Abstract and L43-44 "However, the complexity of this phenomenon causes such models to be either computationally inefficient or unable to handle the overall process.", "The assessment of natural hazards requires predictive numerical models that are able to sufficiently reproduce the studied phenomenon while being efficient in terms of computational resources.". These sentences are quite subjective and are partially related to the previous point raised by the Reviewer. It is certainly true that a predictive numerical tool should be as "computationally efficient" (i.e. fast) as possible. Nevertheless, the first quality for a numerical model, to be considered a predictive tool to assess natural hazard, it should be related to the ability in reproducing adequately the complex phenomena at hand. Thus, the computational efficiency cannot be a strength of the model "per se". On the other hand, a good trade-off between a sufficient/good (but not perfect) reproduction of the phenomenon and a fast computational time is essential when real-time tsunamis early warning systems are considered (e.g. Titov et al., 2005; Cecioni et al., 2011).

-Point-2, Section 3.1.3 The perfectly elastic collision approach, although clearly not correct from a physical point of view, seems to be a clever one for modelling the momentum transfer, at least as a first approach. Nevertheless, few aspects need to be clarified and/or better discussed. First: the Authors claim that the traditional approach for modelling the momentum transfer (e.g. Kelfoun et al., 2010; Xiao et al., 2015) entail undesired user-defined coefficients; on the other hand, also in the proposed approach at least one user-defined calibration coefficient (SF) is needed. Thus, a deeper discussion, as well as more details on the advantages that this approach can bring if compared with the traditional ones, are expected. Second: Figure 2 is not completely clear (and, as a secondary aspect, this Figure has a very poor quality and resolution, please improve it). A discussion of these 2 panels, which likely represent a key aspect of the current approach, is missing in the text. Finally, a more clear description of the Figure (in the legend the Authors refer to "eq. 34", the curves +50 -50 are present in the legend but not described in the caption nor in the text, it is stated that the red line is not the best fitting curve but it is not clear how it has been obtained, etc.) is strongly recommended.

-Point-3, L235 Figures 3 and 4 are poorly described, please improve the description. A brief description of the tsunami generation, well represented by these Figures, can be of interest for the Readers.

-Point-4, L268-272 While describing Figure 5, the Authors point out that "the results from the numerical and physical models are on the same order of magnitude, which permits globally validating the different numerical models but does not allow discrimination between them". Which is certainly true. Nonetheless, one could wonder which parameter, among the landslide thickness (Figure 5a) and the depth-averaged slide velocity (Figure 5b), is more important for the proper momentum transfer modeling. A brief discussion on this would be appropriate.

-Point-5, L344 "We can underline that there is a better match with the runup height than with the wave amplitude" please provide a quantification of the discrepancies between numerical and experimental runup heights and wave amplitudes.

Minor Points:

L20-42: please add some missing references (e.g. Lynett & Liu, 2005; Panizzo et al., 2005; Abadie et al., 2010; Løvholt et al., 2015; Clous & Abadie, 2019)

L59: "wrong", please choose another word or rephrase the whole sentence.

L163: "ultrathin layer of water", please provide more details also considering the option to add a figure with the numerical setup.

Figure 3: Numbers and symbols on the colormaps are too small, please improve it.

Figure 4: please provide more details of the considered numerical simulation in the caption. L299: "more distantly", please change.

References

Abadie, S., Morichon, D., Grilli, S., & Glockner, S. (2010). Numerical simulation of waves generated by landslides using a multiple-fluid Navier-Stokes model. Coastal Engineering, 57 (9), 779–794.

Cecioni, C., Romano, A., Bellotti, G., Di Risio, M., & De Girolamo, P. (2011). Real-time inversion of tsunamis generated by landslides. Natural Hazards & Earth System Sciences, 11 (9).

Clous, L., & Abadie, S. (2019). Simulation of energy transfers 935 in waves generated by granular slides. Landslides, 1–17.

Løvholt, F., Glimsdal, S., Lynett, P., Pedersen, G. (2015). Simulating tsunami propagation in fjords with long-wave models. Natural Hazards & Earth System Sciences, 15 (3).

Lynett, P., & Liu, P. L. F. (2005). A numerical study of the run-up generated by three-dimensional landslides. Journal of Geophysical Research-Oceans, 110 (C3).

Panizzo, A., De Girolamo, P., Di Risio, M., Maistri, A., Petaccia, A. (2005). Great landslide events in Italian artificial reservoirs. Natural Hazards and Earth System Science.

Titov, V. V., Gonzalez, F. I., Bernard, E., Eble, M. C., Mofjeld, H. O., Newman, J. C., & Venturato, A. J. (2005). Real-time tsunami forecasting: Challenges and solutions. Natural Hazards, 35 (1), 35-41.
* * *

---

## Referee Comment (RC2) · Anonymous Referee #2 · 20 Feb 2020

Summary:

The Authors present a numerical study into subaerial granular landslide-tsunamis. The numerical model is a two-layer model based on the shallow-water equations, validated with a range of experiments from Miller et al. (2017). A range of rheological slide models is tested with the Coulumb rheology identified as the best option to model the observed laboratory deposits. The wave amplitudes are investigated at several locations along a flume and compared with the experimental results as well as with predictions based on an empirical equation from the technical literature.

[Figure]

The topic fits well within the scope of NHESS. The approach (elastic collision principle) to model the interface between the slide and water layer is interesting and I can imagine that the Authors invested a lot of time to implement and test the model. Unfortunately, I do not think the selected application (landslide-tsunamis) was a good choice as the applied shallow-water equation model excludes key physical principles for tsunami generation and propagation such as frequency dispersion. This is the reason why shallow-water equation models (in the form used by the Authors) were state-of-the-art 15-10 years ago. The model may be valuable for engineering applications and to speed up the prediction process, however, this is supposed to be a research article. A research article should enhance the physical understanding of a phenomenon. The selected equations by the Authors (the shallow-water equations) are not only outdated for landslide-tsunami modelling, but the Authors also introduce unphysical aspects (e.g. the thin water layer to avoid singularities/numerical issues, the introduction of the fitting parameter in Eq. (31), the unphysical elastic collision principle (L184), the unphysical water surface plot in the last frame of Fig. 4, the back analysis in the Voellmy rheology (258), etc.) and the Authors often fall short in making a precise interpretation of the reason for the mismatch between numerical and experimental results.

In addition, the agreement between numerical and experimental results is rather modest (see e.g. Figs. 9 and 11). Further, I feel that some of the Conclusions contradict the findings in the article (e.g. the Authors praise the good fit of their data in Fig. 11 to the prediction given by Eq. (39) (L417), or they claim that the numerical waves agree well with the experimental one (L410), which is not at all confirmed in Fig. 9, and there is in addition no proper method applied to quantify the numerical-experimental agreement (such as nRMSE)). Strongly related references representing the current state-of-the-art achieving a much better agreement are not includes in the Introduction (e.g. Ma et al. (2015)) and also the presentation of the Figures needs to be significantly improved.

The article requires a lot of work for all these issues to be addressed and to communicate the weaknesses clearer. Below are some more Specific comments together with

a list with Grammar issues and minor points. I recommend a major revision where all these points are carefully addressed.

Specific comments:

A motivation for the Author to use the simplified model is to save computational resources (L44). Therefore, it would be good to discuss this aspect in the manuscript (e.g. what were the used computer resources, what was the computation time of typical run, what is the cell size, etc.)

L19: A landslide can also be partially submerged.

L38: Smoothed Particle Hydrodynamics is a solver, not a set of governing equations. Modelling the generation of subaerial landslide-generated tsunamis with the shallow-water equations was state-of-the-art 15-10 years ago. I feel the way this is written is slightly misleading. Further, more appropriate governing equations for the problem are the RANS equation with turbulence closure, amongst others (LES, DNS, Euler equations).

L41: It is unclear what the Authors mean by "full 3D", Smoothed Particle Hydrodynamics can also address the problem in 3D? Also, coupled approaches are a common trend nowadays, which are not covered, see e.g. Tan et al. (2018). A numerical landslide-tsunami hazard assessment technique applied on hypothetical scenarios at Es Vedrà, offshore Ibiza. Journal of Marine Science and Engineering 6(4):1-22 and other two-layer models such as Ma et al. (2015). A two-layer granular landslide model for tsunami wave generation: Theory and computation. Ocean Modelling 93:40-55. See also the comprehensive review of Yavari-Ramshe and Ataie-Ashtiani (2016). Numerical modeling of subaerial and submarine landslide-generated tsunami waves—recent advances and future challenges. Landslides for a more holistic overview of numerical options.

L41: What is a less classical hybrid approach? Please elaborate.

L43: Why do the predictive methods do need to be numerical, why not physical model studies other methods?

L45: Again, the application of the shallow-water equations for landslide-tsunami generation, at least for subaerial ones, was state-of-the-art around 15 years ago as they exclude, in the general form, key physical concepts such frequency dispersion. It should also be clearer from the text which shallow-water equations are discussed (non-hydrostatic, non-linear, linear)?

L50: It is unclear what the Authors mean by "drag-like equations", please give more details.

L59: "method is obviously wrong from a physical perspective". This is supposed to be a research article, not an Engineering report, so the introduction of physical flawed concepts is slightly questionable. Please comment.

L86: Again, the Authors need to be more precise which shallow-water equations they apply. Some of them are appropriate, at least for tsunami propagation. None of them is appropriate for subaerial landslide-tsunami generation in my view.

L120: The descriptions in the text does not fit with the term "relative density", which indicates one density relative to the other, rather than one density minus another density. The sentence after ("Since each...") does not resolve this confusion well.

L163: I understand that the Authors introduce a small water layer to avoid zeros in the water depth array. In other words, the Authors introduce unphysical boundary conditions to avoid singularities/a numerical issue. This needs a better justification in my view. The fact that this water layer is transferred into a viscous layer does not really resolve this questionable approach. A better option to deal with a numerical issue is on the numerical side, not by compromising the physics.

L184: More unphysical behavior is introduced with the elastic collision principles, research should give new physical insight into phenomena, not to opposite. I invite the

[Figure]

Authors to comment.

Eq. (31): A fitting parameter is introduced. How would the solution look like without fitting parameters? This is a weakness that the maximum slide needs to be known a priori to compute the slide behavior and landslide-tsunami afterwards. Can the Authors discuss where they would know this parameter from in a real scenario?

L226: I am not sure if the Reader is interested in a comparison which is difficult to performed. It looks like this came out of a discussion within the Author team. And it would be better suited in the Discussion section. I suggest removing this paragraph.

The water surface elevation in the last screenshot in Fig. 4 looks rather unphysical.

L250: The Authors aim to predict landslide-tsunamis in nature, by fitting their numerical result to the one in the laboratory. How do the Authors deal with the fact that laboratory granular slide deposits do not represent the behavior in nature at all due to scale effects (see e.g. Kesseler et al. (2020). Grain Reynolds number scale effects in dry granular slides. Journal of Geophysical Research-Earth Surface 125(1):1-19.)? Would their model still be able to predict the phenomenon in nature?

L258: In the Voellmy rheology a back analysis is performed to fit the laboratory experiment. How would this be done if applied to real cases?

L271: "same order of magnitude" means up to a factor of 10 difference. Is this really what the Authors try to state? How useful is a numerical simulation modeling the phenomenon with a difference of up to a factor of 10? Further, in Fig. 5 the mass between the physical and numerical slides seems to be very different. Is the mass at all conserved in the numerical model?

L323: It is unfortunate that the Authors include 5 and 10 cm water depth in their analysis, which are cases in the region of significant scale effects (see the technical literature where h > 0.200 m is specified to avoid significant scale effects for granular slides). The model does not include all physics to represent scale effects (e.g. surface

[Figure]

tension). This is a limitation of the study. E.g. how can the Authors be sure that scale effects are not responsible that their simulation fits well with the experimental results at 5 cm water depth?

Fig. 9: The Authors need to apply a parameter to proper judge the agreement between the laboratory and numerical results, e.g. a normalised Root Mean Square Error (nRMSE) relative to the amplitude, or similar.

Fig. 9: The weaknesses of the shallow-water equation approach become now obvious in the wave profiles, e.g. frequency dispersion is not modelled resulting in significant deviations between numerical and laboratory results. A detailed discussion about frequency dispersion is required, see e.g. Ruffini et al. (2019) Numerical modelling of landslide-tsunami propagation in a wide range of idealised water body geometries. Coastal Engineering 153:103518 or similar studies where landslide-tsunamis are modelled with and without frequency dispersion. The relevance of frequency dispersion on landslide-tsunami propagation is very well known and documented.

L343: The wave train is not simulated because the model is unable to model frequency dispersion. This should be mentioned/discussed.

Figure 11: The data of the numerical simulations are on the un-safe side, which maybe an issue for hazard assessment.

L381: I can not follow why the Authors state that their data are within 30%, it looks rather like a 50% deviation from Eq. (39)?

L384: The fact that the model cannot consider breaking also implies that it cannot consider the impact crater, etc. It would be good to mention/discuss these aspects as well.

L388-293: The argumentation in this section is misleading. The main reason for the absence of the wave train is not breaking, it is the inability of the shallow-water equation (in the form used by the Authors) to model frequency dispersion.

L403: The Conclusions should be understandable on their own. I suggest starting with a brief motivation and method of the study before summarising the conclusions.

L410: I do not agree that this statement is appropriate. The wave in the numerical model behave not similar to the ones in the experiments due to the absence of frequency dispersion in the numerical model. I also do not agree with the conclusion "the choice to transfer the momentum through the simple perfectly elastic collision principle is verified to be relevant." I cannot recall where this has been shown in the article? Please clarify and/or update the Conclusions.

L416: The 211 experiments have been performed by Heller and Hager (2010), 137 ones by Fritz et al. (2002) as well as 86 ones by Zweifel (2004) (see Heller and Hager, 2010).

L417: "...our model is further strengthened by the fact that the results of our model also fit well with those experiments." I disagree, this statement contradicts Fig. 11. The trend of the numerical data is very different to the trend given by Eq. (39), so there is a systematic difference, and the discrepancy is up to 45%.

Suggested grammatical corrections and minor points:

Title: The title includes some repetitions and should be written more concise.

L8: A landslide-generated tsunami is a water wave, it does not involve "landslide dynamics", please rephrase.

L9: Grammar issue, the plural form does not match the singular form used in the previous sentence.

L12: Please write "the shallow water...".

L16: Please write "performs best...".

L19/20: Same issue as on L8.

L28: Landslide-generated tsunamis are not observed on plains, they are observed in the water body. Please rephrase.

L42: Please write "an advanced. . ."

L49: Please follow Harvard style by adding brackets ahead and after the year.

L54: "automatically" is not a good choice, it indicates that the applier needs to do nothing in the simulation.

L57: Consider replacing "specification" with "framework".

L62: Please write "to wave generation".

L65: "Moreover, the granular flow is gravitationally accelerated, which is a relevant aspect to test the behaviour of the numerical model." It is not fully clear what the Authors try to state with this statement, many laboratory slides are accelerated by gravity.

L68: "the comprehensive phenomenon of" is not necessary.

L71: Please replace "but" with "and".

L73: The aluminum plate and the slope are not part of the flume, they are built into the flume, please rephrase.

L75: It is common practice to write parameters in italic style in research articles, so please write $h$ in italic and also all other parameters in the text, tables and figures (e.g. $X$ in Figure 1). On the other hand, numbers (e.g. L152) must not be written in italic.

L79: Please replace "to be" with "as".

L85: Either write "an experiment" or "the experiments".

L91: Please check the numbering style for equations and apply this style consistently throughout the manuscript.

L97: Please check how "Section" is abbreviated (I do not think "sect." is the correct abbreviations) and apply it consistently through the manuscript.

L102: Please follow Harvard Style.

L117: "sin", "cos" and "tan" should not be written in italic, here and throughout the manuscript.

L142: Please add "the" in front of velocity.

L144: The page reference can be dropped, the relation between Chezy and Manning Coefficient is undergraduate student material.

Eq. (16): The writing style of d1/3 seems incorrect, also the number on L154 should be written as subscript.

L151: Please write "stress at which the slide starts or stops to move".

L180/1: Please move "from our point of view" without the brackets after "coefficients".

L206: Please write "Eqs. (29) and (30)", also on L230 and L314.

Figure 2 (and the figures in general): Can the quality of the figures be improved in a professional software such as Adobe Illustrator, etc.? It looks like some of the figures are simple exported from Excel. Also please follow the consistent writing style (e.g. parameters in italic).

L220/1: There are Grammar issues here, the Authors give the impression that Am = 0 in Miller et al. (2017) and it is also unclear what the Authors try to state with "not the best fitting curve". Why?

The Text in Fig. 3 is unreadable small.

Table 1: Please do not use abbreviations, but rather increase the width of the table.

L264: The presentation of the paragraph can also visually be improved, e.g. by intending the first line of the paragraph. Here and throughout the article.

[Figure]

L276: Please write "versus" rather than "vs.".

L281: Please drop "the" in front of best.

P13 at the bottom: Please avoid such free spaces (e.g. by moving the figures), also on P16.

L292: Please drop the in front of sect. and use the correct abbreviation for Section.

Figures 7/8: The text should not be written in italic. Please also write the full term for "w/o"

L315: Please replace "through" with "to".

L316: Please drop one of the 8.

L317: Please write "a result".

L322: Please write "This section investigates the momentum transfer between the slide and the generated wave."

Fig. 9: Again, there are some issues here how the parameters/numbers are written.

Fig. 10: The same issues as elsewhere. Also replace "@" with "at".

L349: I do not think the explanation "The coloured. . ." is needed given that this is visible in the Notation. Similar statements can be dropped in figure captions elsewhere.

L357: Please write "Am is defined. . ."

L364: Please drop "reservoir and" as there was not variation in the reservoir.

Figure 11: Is it possible to better highlight the legend in the figure e.g. by adding a frame? The symbols in the figure may be mixed up with the real data.

L414: Please write "the impulse product parameter particularly. . .", there is only one.

L424: Please write "from all co-authors".

L432: Please write "maximum measured wave amplitude"

L433: Please write "theoretical maximum wave amplitude in the near-field"

Notation, MTs and on the next line: Please use a more formal symbol for "->"

Notation, U: Please write "solution".

References: Please remove inconsistencies such as inconsistent use of upper and lower case letters in the article titles (e.g. L440, L534, etc.), abbreviation and no abbreviation of Journal titles, typos (e.g. L518 "genrated"), etc.

---

## Author Comment (AC1) · 23 Aug 2020

Major Points:

-Point-0, L37-43: As also suggested by Anonymous Referee #2, a more detailed discussion about models based on RANS equations (Reynolds averaged Navier-Stokes equations) will be provided. The enhancement of this discussion will also clarify what we call "full-3D". A more appropriate term will be used. Nevertheless, while this paragraph will be more detailed to better capture the context, we won't present in detail the

pros and cons of each approach.

-Point-1, Abstract and L43-44: The additional discussion based on the RANS equations (Point-0) will present approaches against which our model (or shallow water equations model) can be compared in terms of computational efficiency. This will be also more objective as we will detail more precisely the used computer resources, the computation time of typical run, cell size, etc., as suggested by Anonymous Referee #2. It is true that the efficiency comes second after reproducing the phenomenon adequately but a good compromise between those two constrains is the focus of this study. We think that we have shown in the paper the ability of our model to reproduce physical experiments sufficiently well. We will describe the balance between efficiency and "correctness" with more detail.

Point-2, Section 3.1.3: We will rename our approach as "rigid (discrete) collision" as the term "perfectly elastic" is misleading. Moreover, as only the velocity is exchanged (no change in mass and no deformation), in a very simple way, the approach remains very simple. First: The "traditional approach", i.e. transferring the momentum through drag forces, is in fact applied using different sets of equations (or different level of complexity) in literature, thus not so traditional. Moreover, it is relatively not well suited in free surface models. Concerning the undesired user-defined coefficient, the number of coefficient is often relatively important. For instance, Kelfoun, K., Giachetti, T. & Lazabazuy, P. (2010). Landslide-genrated tsunami at Téunion Island. J. Geophys. Res., 115. : 2 coefficients ; Xiao, L., Ward, S. & Wang, J. (2015). Tsunami Squares Approach to Landslide-Generated Waves: Application to Gongjiafang Landslide, Three Gorges Reservoir, China. Pure Appl. Geophys., 172, 3639-3654. : 2 coefficients. Our model indeed requires some manual adjustment through the FS, but it should be performed automatically in a close future. Second: The figure 2 will be improved. And a more advanced discussion will be provided. Finally: The description of the figure will be improved.

Point-3, L235: The figures 3 and 4 will be better described. The description of the

generation will be added.

Point 4, L268-272: we will add discussion about the effects of the velocity and the landslide thickness on the momentum transfer through a sensibility analysis for instance.

Point-5, L344: We agree that the discussion about the run-up was relatively poor. We will discuss more precisely the results and quantify of the discrepancies together with a discussion about the potential causes of these problems.

Minor points: All the suggested corrections and Improvements will be done.

―――――――――――――――――

---

## Author Comment (AC2) · 23 Aug 2020

Specific comments:

We agree with the statement that it would be appropriate to discuss the used computer resources, the computation time of typical run, the cell size etc. These aspects will be presented in the section 3.

L19(L22): We will add "partially submerged".

[Figure]

L38 & L41: As answered to anonymous referee #1, a more detailed discussion about models based on RANS equations (Reynolds averaged Navier-Stokes equations) will be provided. The enhancement of this discussion will also clarify what we call "full-3D" or "less classical hybrid approach (which will be rephrased)". A more appropriate term will be used. Nevertheless, while this paragraph will be more detailed to better capture the context, we won't present in detail the pros and cons of each approach.

L43: Indeed, predictive model can be of different nature (as described in the paragraph above). This sentence will be rewritten more clearly.

L 50: We will add a sentence that explain the approaches using drag equations. The "drag-like equation" term was used because there are different approach and simplification used in the literature that make not all of them fall under the definition.

L59: We will rename our approach as "rigid (discrete) collision" as the term "perfectly elastic" is misleading. Moreover, as only the velocity is exchanged (no change in mass and no deformation), in a very simple way, the approach remains very simple and is, in fact, correct in a physical point of view. The paragraph will be rewritten.

L86: The non-linear shallow water equations is in our point of view appropriate as it works well for tsunami propagation (as mentioned by Referee #2) and landslide propagation (as demonstrated in the manuscript and in literature, e. g. Hungr and Evans, 1996). The interaction between these two layers is the core of the paper and we think it performed well, as demonstrated through the effect of the momentum transfer on the landslide deposit and the generated waves.

L120: It is true that  cannot be named "relative density" in this context. We will rename it with a new term such as "buoyant density". The concept of relative density appears when it is divided by s.

L163: We will provide a more detailed explanation in the final paper. This is justified because using this approach, there is no special need to introduce new equations for

wet-dry transition.

L184: We will rename our approach as "rigid (discrete) collision" as the term "perfectly elastic" is misleading. Moreover, as only the velocity is exchanged (no change in mass and no deformation), in a very simple way, the approach remains very simple and is, in fact, correct in a physical point of view. Eq. (31): In fact, the maximum slide thickness is not a manual input. During the simulation, the landslide maximum thickness is tracked (and registered) in the numerical equivalent location of the Cam1 (Figure 1). In the case of a real scenario, the same principle will apply. The operator only need to point out where this value is numerically measured.

L226: It is sure that this paragraph is not strategically placed in the manuscript. We will either integrate it in the discussion or remove it.

L250: In this paper, we present the comparison between our model and a specific physical experiment. It shows that when we implement exactly the measured rheological parameters in our model (for granular flow), the behaviour of the landslide is correct. Moreover, speaking about the dry case, our model was by no mean modified to fit the physical experiment. The isotropic Coulomb rheology was used and the friction angles were implemented, that's it. Thus, we think that it is a robust approach to validate a numerical model. While we won't discuss the scale effects on the physical experiment conducted in Miller et al. (2017) here, the fact that they used slide masses greater than 500 kg (while Kesseler et al. (2020) investigated side masses between 1 and 110 kg), place this particular study on the "positive" side. Back analysis of real cases are also a way to validate numerical models, but "playing" with the input parameters and tuning the code was not the topic of this paper. We have already used our model in real case study, either prospective or in back analysis, and showed good capacities. Unfortunately, at this time, the code was not in its actual state (e. g. the momentum transfer was not implemented). (e. g. Franz, M., Rudaz, B., Jaboyedoff, M. & Podladchikov, Y. (2016). Fast assessment of landslide-generated tsunami and associated risks by coupling SLBL with shallow water model. Proceeding of the GEOVancouver

2016 conference; Franz, M., Jaboyedoff, J., Podladchikov, Y. & Locat, J. (2015). Testing a landslide-generated tsunami model. The case of the Nicolet Landslide (Québec, Canada). Proceeding of the GEOQuébec 2015 conference). The use of the present state model in real case scenarios will be presented in a future paper to address this crucial point, on which we agree.

L258: The Voellmy rheology is widely used to simulate snow and rock avalanches. The parameters used in prospective study are often based on regional back analysis cases (e. g. Hungr, O. & Evans, S.G. (1996). Rock avalanche runout prediction using dynamic model. Landslides.).

L271: a) We won't use the "same order of magnitude" expression. b) Yes, the mass is conserved in the numerical model. The difference in fig. 5 are due to an incomplete graph (stops at 3.2 s) and the fact that thickness of the landslide in the physical experiment is expended (bouncing beads). This is not reproduced numerically.

L323: We think that the scale effect is not dominant in this case. This is highlighted by the fact that fairly good correspondence between the laboratory and the numerical simulation are present over and under the 0.2 m threshold Referee #2 mentions.

Fig. 9: The suggestion to use the nRMSE to judge the agreement between laboratory and numerical results is welcome. We will apply that. Fig. 9: We don't think that the lack of frequency dispersion is a complete weakness of the approach. Nevertheless, we will add a detailed discussion on the topic.

L 343: The link between the wave trains and the frequency dispersion will be discussed.

Figure 11: The data of the numerical simulations are indeed a little lower (smaller Am) than the lab data and it would have been beneficial for hazard assessment to be in the opposite case. Nevertheless, they stay within the +/-30% deviation from the Eq. (39) which is considered as a good fit (Heller et al., 2010).

L381: The dashed lines are strictly the Eq. (39) +/- 30%. It is obvious, for instance,

if we check the plain line at Am = 3. The upper dashed line is at Am = 4 (3+(3*0.3)) and the lower dashed line is at Am = 2 (3-(3*0.3). Therefore, as the data are within the dashed lines (except for 0.05m) they are within +/- 30%. The maximum relative wave amplitude Am is a function of the impulse product parameter P. We also should keep in mind that the figure 11 uses the same axis lengths as in Miller et al. (2017). The figure presented in Heller et Hager (2010) has the P axis that stops at 9, which means that the relationship from Eq. (39) is not valid after this point. Nevertheless, the numerical data for a water depth of 0.05 m is not so distant from the laboratory data.

L384: While it is true that the model cannot handle breaking, it is not true that it cannot handle the impact crater, as long as the steepness of the water surface is not steeper than "sub-vertical". We will mention this aspect in the final paper.

L388-293: The argumentation will be corrected, in parallel with the correction regarding Fig. 9.

L403: We will begin the chapter with a paragraph concerning the motivation and the methods of the study.

L410: We will clarify and update the Conclusion. This will be link with all the enhancement throughout the manuscript.

L416: This will be corrected.

L 417: As discussed for the L381, if we look at the data for a P smaller than 9, the fit is good. Moreover, looking at the figure depicting Am and P relationship in Heller and Hager (2010), we see that the data are contained between the +/- 30 % lines. It is the case of the data from our numerical model. Except for h=0.05m, which is anyway for a P greater than 9. Suggested grammatical corrections and minor points:

L220/1: We will rephrase. Indeed, Am is not equal to 0 in Miller et al. (2017). Here we plot the deviation from Am in Miller et al. (2017). We mean: "The value used in Eq. (33) (red line) is not the best fitting curve" in the panel a). But they are the best

compromise for a) AND b).

All the other suggested corrections and improvements will be done.

---

## Author Response (AR1)

**Author's response to anonymous referee #1**

Major Points:

-Point-0, L37-43 A more detailed discussion of more complex and time consuming numerical models (based on the RANS equations, e.g. Abadie et al., 2010; Clous & Abadie, 2019) is needed. This would help the Readers in comparing the approach proposed by the Authors with the ones available in the scientific literature (see also Point-1).

Answer:

- RANS equations have been mentioned and the suggested references have been added. The paragraph has been rewritten(L33-55).
- Concerning the computation time, direct comparison is not feasible, it depends on the spatial and temporal resolution. We add a section about this topic (L400-403).

-Point-1, Abstract and L43-44 "However, the complexity of this phenomenon causes such models to be either computationally inefficient or unable to handle the overall process.", "The assessment of natural hazards requires predictive numerical models that are able to sufficiently reproduce the studied phenomenon while being efficient in terms of computational resources.". These sentences are quite subjective and are partially related to the previous point raised by the Reviewer. It is certainly true that a predictive numerical tool should be as "computationally efficient" (i.e. fast) as possible. Nevertheless, the first quality for a numerical model, to be considered a predictive tool to assess natural hazard, it should be related to the ability in reproducing adequately the complex phenomena at hand. Thus, the computational efficiency cannot be a strength of the model "per se". On the other hand, a good trade-off between a sufficient/good (but not perfect) reproduction of the phenomenon and a fast computational time is essential when real-time tsunamis early warning systems are considered (e.g. Titov et al., 2005; Cecioni et al., 2011).

Answer:

- The paragraph has been rewritten (L34-68). Nevertheless, in general, the comments of both reviewers show that they understand the statement of the balance "correctness-efficiency" and that we had to make a choice of the level of approximation. We chose the non-linear shallow water equations (without multiple add-ons), hence, the inherent approximations. We won't discuss in the paper the well-known limitations of this approach.

-Point-2, Section 3.1.3 The perfectly elastic collision approach, although clearly not correct from a physical point of view, seems to be a clever one for modelling the momentum transfer, at least as a first approach. Nevertheless, few aspects need to be clarified and/or better discussed. First: the Authors claim that the traditional approach for modelling the momentum transfer (e.g. Kelfoun et al., 2010; Xiao et al., 2015) entail undesired user-defined coefficients; on the other hand, also in the proposed approach at least one user-defined calibration coefficient (SF) is needed. Thus, a deeper discussion, as well as more details on the advantages that this approach can bring if compared with the traditional ones, are expected. Second: Figure 2 is not completely clear (and, as a secondary aspect, this Figure has a very poor quality and resolution, please improve it). A discussion of these 2 panels, which likely represent a key aspect of the current approach, is missing in the text. Finally, a more clear description of the Figure (in the legend the Authors refer to "eq. 34", the curves +50 -50 are present

in the legend but not described in the caption nor in the text, it is stated that the red line is not the best fitting curve but it is not clear how it has been obtained, etc.) is strongly recommended.

Answer:

- First: The description of the approach of Kelfoun et al. (2010) and Xiao et al. (2015) have been improved (L60-66).
- Second: The figure 2 has been updated (quality, legend) and the description has been improved (Fig.2; L223-230).

-Point-3, L235 Figures 3 and 4 are poorly described, please improve the description. A brief description of the tsunami generation, well represented by these Figures, can be of interest for the Readers.

Answer:

- The figure 3 has been improved and a more detailed description has been done (Fig. 3; L246-258)

-Point-4, L268-272 While describing Figure 5, the Authors point out that "the results from the numerical and physical models are on the same order of magnitude, which permits globally validating the different numerical models but does not allow discrimination between them". Which is certainly true. Nonetheless, one could wonder which parameter, among the landslide thickness (Figure 5a) and the depth-averaged slide velocity (Figure 5b), is more important for the proper momentum transfer modeling. A brief discussion on this would be appropriate.

Answer:

- While this suggestion is relevant, the investigation we performed to determine, in this particular case, which parameter has more influence on the wave amplitude was not applicable. Indeed, the thickness and the velocity of the landslide is directly linked to the parameters of the slide material and the apparatus. The realisation of this analysis would have required to simulate new scenarios, which is beyond the scope of this study.

-Point-5, L344 "We can underline that there is a better match with the runup height than with the wave amplitude" please provide a quantification of the discrepancies between numerical and experimental runup heights and wave amplitudes.

Answer:

- -we decided to remove the section concerning the runup, as it is a bit out of scope of the aim of the study and because of inconsistencies.

Minor Points:

L20-42: please add some missing references (e.g. Lynett & Liu, 2005; Panizzo et al., 2005; Abadie et al., 2010; Løvholt et al., 2015; Clous & Abadie, 2019):

- Some references added (L33-68)

L59: "wrong", please choose another word or rephrase the whole sentence:

- Changed (L73)

L163: "ultrathin layer of water", please provide more details also considering the option to add a figure with the numerical setup:

- We think a development of this concept isn't relevant.

Figure 3: Numbers and symbols on the colormaps are too small, please improve it:

- Figure Improved (Fig. 3)

Figure 4: please provide more details of the considered numerical simulation in the caption:

- More detail in the text (L254-258). We won't add more detail as the idea of this figure was to illustrate the cross-section of the 2.5D model.

L299: "more distantly", please change.

- Changed (L316)

**Author's response to anonymous referee #2**

Specific comments:

A motivation for the Author to use the simplified model is to save computational resources (L44). Therefore, it would be good to discuss this aspect in the manuscript (e.g. what were the used computer resources, what was the computation time of typical run, what is the cell size, etc.).

Answer:

- We add a section about computational resources (L400-403).

L19: A landslide can also be partially submerged.

Answer:

- Corrected (L23-24)

L38: Smoothed Particle Hydrodynamics is a solver, not a set of governing equations. Modelling the generation of subaerial landslide-generated tsunamis with the shallow water equations was state-of-the-art 15-10 years ago. I feel the way this is written is slightly misleading. Further, more appropriate governing equations for the problem are the RANS equation with turbulence closure, amongst others (LES, DNS, Euler equations).

Answer:

- RANS equations have been mentioned. The paragraph has been rewritten (L33-55).

L41: It is unclear what the Authors mean by "full 3D", Smoothed Particle Hydrodynamics can also address the problem in 3D? Also, coupled approaches are a common trend nowadays, which are not covered, see e.g. Tan et al. (2018). A numerical landslidetsunami hazard assessment technique applied on hypothetical scenarios at Es Vedrà, offshore Ibiza. Journal of Marine Science and Engineering 6(4):1-22 and other two-layer models such as Ma et al. (2015). A two-layer granular landslide model for tsunami wave generation: Theory and computation. Ocean Modelling 93:40-55. See also the comprehensive review of Yavari-Ramshe and Ataie-Ashtiani (2016). Numerical modeling of subaerial and submarine landslide-generated tsunami wavesˇA ˇTrecent advances and future challenges. Landslides for a more holistic overview of numerical options.

L41: What is a less classical hybrid approach? Please elaborate.

Answer:

- Coupled approaches have been mentioned and some references added (L33-55).
- The paragraph has been rewritten (L33-55).

L43: Why do the predictive methods do need to be numerical, why not physical model studies other methods?

Answer:

- They don't. Sentence rewritten (L45)

L45: Again, the application of the shallow-water equations for landslide-tsunami generation, at least for subaerial ones, was state-of-the-art around 15 years ago as they exclude, in the general form, key physical concepts such frequency dispersion. It should also be clearer from the text which shallow-water equations are discussed (nonhydrostatic, non-linear, linear)?

Answer:

- The comments of both reviewers show that they understand the statement of the balance "correctness-efficiency" and that we had to make a choice of the level of approximation. We chose the non-linear shallow water equations (without multiple add-ons), hence, the inherent approximations and incomplete physics (such as frequency dispersion). We won't discuss in the paper the well-known limitations of this approach.
- Some precision concerning the shallow water equations have been added (L70, L100).

L50: It is unclear what the Authors mean by "drag-like equations", please give more details.

Answer:

- The paragraph has been enhanced (L60-68).

L59: "method is obviously wrong from a physical perspective". This is supposed to be a research article, not an Engineering report, so the introduction of physical flawed concepts is slightly questionable. Please comment.

Answer:

- We have renamed our approach as "perfect" collision as the term "perfectly elastic" was misleading. It is "perfect" in the sense that there is no dissipation, neither the momentum nor the kinetic energy.
- Rewritten in the text (L71-72).

L86: Again, the Authors need to be more precise which shallow-water equations they apply. Some of them are appropriate, at least for tsunami propagation. None of them is appropriate for subaerial landslide-tsunami generation in my view.

Answer:

- Some precision concerning the shallow water equations have been added (L70, L100).
- The comments of both reviewers show that they understand the statement of the balance "correctness-efficiency" and that we had to make a choice of the level of approximation. We chose the non-linear shallow water equations (without multiple add-ons), hence, the inherent approximations and incomplete physics (such as frequency dispersion). We won't discuss in the paper the well-known limitations of this approach.

L120: The descriptions in the text does not fit with the term "relative density", which indicates one density relative to the other, rather than one density minus another density. The sentence after ("Since each…") does not resolve this confusion well.

Answer:

- The term has been changed to "apparent density" (L133 and Notation).

L163: I understand that the Authors introduce a small water layer to avoid zeros in the water depth array. In other words, the Authors introduce unphysical boundary conditions to avoid singularities/a numerical issue. This needs a better justification in my view. The fact that this water layer is transferred into a viscous layer does not really resolve this questionable approach. A better option to deal with a numerical issue is on the numerical side, not by compromising the physics.

Answer:

- This is justified because using this approach, there is no special need to introduce new equations for wet-dry transition.

L184: More unphysical behavior is introduced with the elastic collision principles, research should give new physical insight into phenomena, not to opposite. I invite the Authors to comment.

Answer:

- We have renamed our approach as "perfect" collision as the term "perfectly elastic" was misleading. It is "perfect" in the sense that there is no dissipation, neither the momentum nor the kinetic energy.

Eq. (31): A fitting parameter is introduced. How would the solution look like without fitting parameters? This is a weakness that the maximum slide needs to be known a priori to compute the slide behavior and landslide-tsunami afterwards. Can the Authors discuss where they would know this parameter from in a real scenario?

Answer:

- In fact, the maximum slide thickness is not a manual input. During the simulation, the landslide maximum thickness is tracked (and registered) in the numerical equivalent location of the Cam1 (Figure 1). In the case of a real scenario, the same principle will apply. The operator only need to point out where this value is numerically measured.
- This aspect has been precised in the text (L225).

L226: I am not sure if the Reader is interested in a comparison which is difficult to performed. It looks like this came out of a discussion within the Author team. And it would be better suited in the Discussion section. I suggest removing this paragraph.

Answer:

- Paragraph removed (L238-244)

The water surface elevation in the last screenshot in Fig. 4 looks rather unphysical.

Answer:

- This is due to the approximation inherent to the shallow water equations.

L250: The Authors aim to predict landslide-tsunamis in nature, by fitting their numerical result to the one in the laboratory. How do the Authors deal with the fact that laboratory granular slide deposits do not represent the behavior in nature at all due to scale effects (see e.g. Kesseler et al. (2020). Grain Reynolds number scale effects in dry granular slides. Journal of Geophysical Research-Earth Surface 125(1):1-19.)? Would their model still be able to predict the phenomenon in nature?

Answer:

- In this paper, we present the comparison between our model and a specific physical experiment. It shows that when we implement exactly the measured rheological parameters in our model (for granular flow), the behaviour of the landslide is correct. Moreover, speaking about the dry case, our model was by no mean modified to fit the physical experiment. The isotropic Coulomb rheology was used and the friction angles were implemented, that's it. Thus, we think that it is a robust approach to validate a numerical model.
- While we won't discuss the scale effects on the physical experiment conducted in Miller et al. (2017) here, the fact that they used slide masses greater than 500 kg (while Kesseler et al. (2020) investigated side masses between 1 and 110 kg), place this particular study on the "positive" side.
- Back analysis of real cases are also a way to validate numerical models, but "playing" with the input parameters and tuning the code was not the topic of this paper. We have already used our model in real case study, either prospective or in back analysis, and showed good capacities. Unfortunately, at this time, the code was not in its actual state (e. g. the momentum transfer was not implemented). (e. g. Franz, M., Rudaz, B., Jaboyedoff, M. & Podladchikov, Y. (2016). Fast assessment of landslide-generated tsunami and associated risks by coupling SLBL with shallow water model. Proceeding of the GEOVancouver 2016 conference; Franz, M., Jaboyedoff, J., Podladchikov, Y. & Locat, J. (2015). Testing a landslide-generated tsunami model. The case of the Nicolet Landslide (Québec, Canada). Proceeding of the GEOQuébec 2015 conference). The use of the present state model in real case scenarios will be presented in a future paper to address this crucial point, on which we agree.

L258: In the Voellmy rheology a back analysis is performed to fit the laboratory experiment. How would this be done if applied to real cases?

Answer:

- The Voellmy rheology is widely used to simulate snow and rock avalanches. The parameters used in prospective study are often based on regional back analysis cases (e. g. Hungr, O. & Evans, S.G. (1996). Rock avalanche runout prediction using dynamic model. Landslides.).

L271: "same order of magnitude" means up to a factor of 10 difference. Is this really what the Authors try to state? How useful is a numerical simulation modeling the phenomenon with a difference of up to a factor of 10? Further, in Fig. 5 the mass between the physical and numerical slides seems to be very different. Is the mass at all conserved in the numerical model?

Answer:

- "same order of magnitude" has been changed in the text (L288-290).
- Yes, the mass is conserved in the numerical model. The difference in fig. 5 are due to an incomplete graph (stops at 3.2 s) and the fact that thickness of the landslide in the physical experiment is expended (bouncing beads). This is not reproduced numerically.

L323: It is unfortunate that the Authors include 5 and 10 cm water depth in their analysis, which are cases in the region of significant scale effects (see the technical literature where h > 0.200 m is specified to avoid significant scale effects for granular slides). The model does not include all physics to represent scale effects (e.g. surface tension). This is a limitation of the study. E.g. how can the Authors be sure that scale effects are not responsible that their simulation fits well with the experimental results at 5 cm water depth?

Answer:

- We think that the scale effect is not dominant in this case. This is highlighted by the fact that fairly good correspondence between the laboratory and the numerical simulation are present over and under the 0.2 m threshold Referee #2 mentions.

Fig. 9: The Authors need to apply a parameter to proper judge the agreement between the laboratory and numerical results, e.g. a normalised Root Mean Square Error (nRMSE) relative to the amplitude, or similar.

Answer:

- The nRMSE is too dependent to the location in time of the wave or to the size of the windows in which it would be computed. We will not perform it.

Fig. 9: The weaknesses of the shallow-water equation approach become now obvious in the wave profiles, e.g. frequency dispersion is not modelled resulting in significant deviations between numerical and laboratory results. A detailed discussion about frequency dispersion is required, see e.g. Ruffini et al. (2019) Numerical modelling of landslide-tsunami propagation in a wide range of idealised water body geometries. Coastal Engineering 153:103518 or similar studies where landslide-tsunamis are modelled with and without frequency dispersion. The relevance of frequency dispersion on landslide-tsunami propagation is very well known and documented.

Answer:

- The comments of both reviewers show that they understand the statement of the balance "correctness-efficiency" and that we had to make a choice of the level of approximation. We chose the non-linear shallow water equations (without multiple add-ons), hence, the inherent

approximations and incomplete physics (such as frequency dispersion). We won't discuss in the paper the well-known limitations of this approach.

L343: The wave train is not simulated because the model is unable to model frequency dispersion. This should be mentioned/discussed.

Answer:

- Also due to the lack of breaking wave in the model. Mentioned in the text and paragraph rewritten (L409-415).

Figure 11 (now Fig. 10): The data of the numerical simulations are on the un-safe side, which maybe an issue for hazard assessment.

Answer:

- The data of the numerical simulations are indeed a little lower (smaller Am) than the lab data and it would have been beneficial for hazard assessment to be in the opposite case. Nevertheless, they stay within the +/-30% deviation from the Eq. (39) which is considered as a good fit (Heller et al., 2010).

L381: I cannot follow why the Authors state that their data are within 30%, it looks rather like a 50% deviation from Eq. (39)?

Answer:

- The dashed lines are strictly the Eq. (39) +/- 30%. It is obvious, for instance, if we check the plain line at $Am$ = 3. The upper dashed line is at $Am$ = 4 (3+(3*0.3)) and the lower dashed line is at $Am$ = 2 (3-(3*0.3). Therefore, as the data are within the dashed lines (except for 0.05m) they are within +/- 30%. The maximum relative wave amplitude $Am$ is a function of the impulse product parameter P. We also should keep in mind that the figure 11 uses the same axis lengths as in Miller et al. (2017). The figure presented in Heller et Hager (2010) has the P axis that stops at 9, which means that the relationship from Eq. (39) is not valid after this point. Nevertheless, the numerical data for a water depth of 0.05 m is not so distant from the laboratory data.

L384: The fact that the model cannot consider breaking also implies that it cannot consider the impact crater, etc. It would be good to mention/discuss these aspects as well.

Answer:

- While it is true that the model cannot handle breaking, it is not true that it cannot handle the impact crater, as long as the steepness of the water surface is not steeper than "sub-vertical".
- Paragraph updated (L405-409).

L388-293: The argumentation in this section is misleading. The main reason for the absence of the wave train is not breaking, it is the inability of the shallow-water equation (in the form used by the Authors) to model frequency dispersion.

Answer:

- Also due to the lack of breaking wave in the model. Mentioned in the text and paragraph rewritten (L409-415).

L403: The Conclusions should be understandable on their own. I suggest starting with a brief motivation and method of the study before summarising the conclusions.

Answer:

- A new paragraph has been added at the begging of the section (L428-432).

L410: I do not agree that this statement is appropriate. The wave in the numerical model behave not similar to the ones in the experiments due to the absence of frequency dispersion in the numerical model. I also do not agree with the conclusion "the choice to transfer the momentum through the simple perfectly elastic collision principle is verified to be relevant." I cannot recall where this has been shown in the article? Please clarify and/or update the Conclusions.

Answer:

- The comments of both reviewers show that they understand the statement of the balance "correctness-efficiency" and that we had to make a choice of the level of approximation. We chose the non-linear shallow water equations (without multiple add-ons), hence, the inherent approximations and incomplete physics (such as frequency dispersion). We won't discuss in the paper the well-known limitations of this approach.
- We consider the results sufficiently good from this point of view.

L416: The 211 experiments have been performed by Heller and Hager (2010), 137 ones by Fritz et al. (2002) as well as 86 ones by Zweifel (2004) (see Heller and Hager, 2010).

Answer:

- This has been corrected (L445-446)

L417: "… our model is further strengthened by the fact that the results of our model also fit well with those experiments." I disagree, this statement contradicts Fig. 11. The trend of the numerical data is very different to the trend given by Eq. (39), so there is a systematic difference, and the discrepancy is up to 45%.

Answer:

- The dashed lines in Figure 10 are strictly the Eq. (39) +/- 30%. It is obvious, for instance, if we check the plain line at $Am$ = 3. The upper dashed line is at $Am$ = 4 (3+(3*0.3)) and the lower dashed line is at $Am$ = 2 (3-(3*0.3). Therefore, as the data are within the dashed lines (except for 0.05m) they are within +/- 30%. The maximum relative wave amplitude $Am$ is a function of the impulse product parameter P. We also should keep in mind that the figure 11 uses the same axis lengths as in Miller et al. (2017). The figure presented in Heller et Hager (2010) has the P axis that stops at 9, which means that the relationship from Eq. (39) is not valid after this point. Nevertheless, the numerical data for a water depth of 0.05 m is not so distant from the laboratory data.

Suggested grammatical corrections and minor points:

Title: The title includes some repetitions and should be written more concise:

- We keep it as it is.

L8: A landslide-generated tsunami is a water wave, it does not involve "landslide dynamics", please rephrase;

L9: Grammar issue, the plural form does not match the singular form used in the previous sentence.

- The sentence has been rewritten (L8-10)

L12: Please write "the shallow water…":

- Done (L13)

L16: Please write "performs best…":

- Done (L17)

L19/20: Same issue as on L8:

- Done (L22)

L28: Landslide-generated tsunamis are not observed on plains, they are observed in the water body. Please rephrase:

- Rephrased (L30)

L42: Please write "an advanced…":

- Done (L54)

L49: Please follow Harvard style by adding brackets ahead and after the year.

- Done (L60)

L54: "automatically" is not a good choice, it indicates that the applier needs to do nothing in the simulation.

- Deleted (L67)

L57: Consider replacing "specification" with "framework":

- Done (L71)

L62: Please write "to wave generation".

- Done (L76-77)

L65: "Moreover, the granular flow is gravitationally accelerated, which is a relevant aspect to test the behaviour of the numerical model." It is not fully clear what the Authors try to state with this statement, many laboratory slides are accelerated by gravity.

- Yes, that's true. We chose one of them, which in this study, is the one developed by Miller et al. (2017).

L68: "the comprehensive phenomenon of" is not necessary:

- Deleted (L83)

L71: Please replace "but" with "and":

- Done (L86)

L73: The aluminium plate and the slope are not part of the flume, they are built into the flume, please rephrase:

- Rewritten (L88)

L75: It is common practice to write parameters in italic style in research articles, so please write h in italic and also all other parameters in the text, tables and figures (e.g. X in Figure 1). On the other hand, numbers (e.g. L152) must not be written in italic:

- It has been changed in the whole document.

L79: Please replace "to be" with "as":

- Don (L94)

L85: Either write "an experiment" or "the experiments":

- Done (L99)

L91: Please check the numbering style for equations and apply this style consistently throughout the manuscript:

- It has been corrected in the whole document.7

L97: Please check how "Section" is abbreviated (I do not think "sect." is the correct abbreviations) and apply it consistently through the manuscript:

- Done (L111, L309)

L102: Please follow Harvard Style:

- Done (L116)

L117: "sin", "cos" and "tan" should not be written in italic, here and throughout the manuscript:

- Done

L142: Please add "the" in front of velocity:

- Done (L153)

L144: The page reference can be dropped, the relation between Chezy and Manning Coefficient is undergraduate student material:

- Page reference dropped (L155)

Eq. (16): The writing style of d1/3 seems incorrect, also the number on L154 should be written as subscript.

- Style changed (L160)

L151: Please write "stress at which the slide starts or stops to move":

- Changed (L162-163)

L180/1: Please move "from our point of view" without the brackets after "coefficients":

- Corrected (L190-191)

L206: Please write "Eqs. (29) and (30)", also on L230 and L314:

- Done (L217, 331)

Figure 2 (and the figures in general): Can the quality of the figures be improved in a professional software such as Adobe Illustrator, etc.? It looks like some of the figures are simple exported from Excel. Also please follow the consistent writing style (e.g. parameters in italic):

- The figures have been improved and the writing style has been harmonised (all figures)

L220/1: There are Grammar issues here, the Authors give the impression that Am = 0 in Miller et al. (2017) and it is also unclear what the Authors try to state with "not the best fitting curve". Why?

- The legend has been improved as well as the text and the caption. (Fig. 2; L227-235)
- It is not the best fitting curve regarding only phenomenon, but it is the best for landslide and water.

The Text in Fig. 3 is unreadable small:

- Figure enhanced (Fig. 3)

Table 1: Please do not use abbreviations, but rather increase the width of the table:

- Full words written (Tab. 1)

L264: The presentation of the paragraph can also visually be improved, e.g. by intending the first line of the paragraph. Here and throughout the article:

- We have followed the guidelines

L276: Please write "versus" rather than "vs.":

- Done (L294)

L281: Please drop "the" in front of best:

- Dropped (L296)

P13 at the bottom: Please avoid such free spaces (e.g. by moving the figures), also on P16:

- This will be performed for the very last version of the manuscript

L292: Please drop the in front of sect. and use the correct abbreviation for Section:

- Done (L309)

Figures 7/8: The text should not be written in italic. Please also write the full term for "w/o":

- Done (Fig.7, Fig.8)

L315: Please replace "through" with "to":

- Done (L332)

L316: Please drop one of the 8:

- Done (L333)

L317: Please write "a result":

- Wrote "the" (L333)

L322: Please write "This section investigates the momentum transfer between the slide and the generated wave.":

- Done (L339)

Fig. 9: Again, there are some issues here how the parameters/numbers are written.

- Corrected (Fig.9)

Fig. 10: The same issues as elsewhere. Also replace "@" with "at" :

- Figure deleted

L349: I do not think the explanation "The coloured: : :" is needed given that this is visible in the Notation. Similar statements can be dropped in figure captions elsewhere:

- This particular caption has been removed.
- The unneeded explanations have been removed from the other figures' captions (All Figs.)

L357: Please write "Am is defined…":

- Done (L374)

L364: Please drop "reservoir and" as there was not variation in the reservoir:

- Done (L381)

Figure 11 (Now Fig. 10): Is it possible to better highlight the legend in the figure e.g. by adding a frame? The symbols in the figure may be mixed up with the real data:

- A frame has been added (Fig. 10)

L414: Please write "the impulse product parameter particularly…", there is only one:

- Corrected (L444)

L424: Please write "from all co-authors":

- Done (L455)

Notations:

- Done

References: Please remove inconsistencies such as inconsistent use of upper and lower case letters in the article titles (e.g. :L440, L534, etc.), abbreviation and no abbreviation of Journal titles, typos (e.g. L518 "genrated"), etc.:

- Done

[revised manuscript text omitted]

---

## Author Response (AR2)

**Author's response to anonymous referee #2**

Suggestion for revision:

**Summary:**

The Authors improved many aspects in the manuscript and the modelling of the slide phase includes some advancements with the limitations (e.g. the shape factor as a fitting parameter) clearly communicated. However, the modelling of the landslide-tsunamis is a step backward (shallow-water equations (SWEs), excluding frequency dispersion) in my view, reflecting the state of the art 10-15 years ago. The Authors do not fully acknowledge these limitations in the article, despite the concerns of both Reviewers. Yes, there is a new statement on L368 about these limitations, but these limitations also need to be made clear in other key sections such as the Introduction, Conclusions, during the discussions of Figs. 4 and 9 (where the limitations of the applied model are obvious), etc. Further, there are still some linguistic issues, typos and smaller inconsistencies. Finally, as already highlighted in the 1st review, I do not think that some key conclusions are supported by the findings in the article. More details are given below.

- Authors: the limitations have been developed in the respective sections

**Specific comments:**

L39: Good that the RANS equations are now introduced as well as an alternative to the model applied by the Authors, but the full term needs to be introduced, not only the abbreviation. More background into other options (DNS, LES) would also be desirable.

- Authors: the full terms have been introduced and the other suggested options have been included (L40-41).

L46/L185: It is not clear what the Authors mean by "free-surface nature". Do not all mentioned models have a free surface? Or do they mean the way the free surface is treated (e.g. Volume of Fluids)?

- Authors: The terms have been changed (L51, 54, 64, 199)

L315 and Fig. 9: "…the numerical simulation reproduces the wave observed in the physical experiment very well in terms of amplitude and timing at each probe." The "very well" is subjective and needs to be backed up with a more objective criterion (goodness of fit parameter). It is fine if the Authors do not want to use the nRMSE, but they need to use an alternative way to scientifically quantify the goodness of fit (e.g. a % deviations between the amplitudes), as already pointed out by both Reviewers in the 1st review round in other contexts.

- Authors: % of deviation have been added (L339-349)

Fig. 9 (first panel): The numerical wave appears to be cut? This makes it impossible to appreciate the quality of the agreement.

- Authors: The top of the wave have been added (Fig. 9)

L359: This new paragraph is helpful showing that a simulation is performed efficiently. However, I believe this section is at the incorrect place in the manuscript (it should be part of the Methodology). Further, have convergence tests been performed?

- Authors: The paragraph has been moved (Sect. 3.2). No convergence tests have been performed.

L396: "…overall, the model effectively handles the complex phenomenon occurring during the interaction between the landslide and the water." I do not fully agree with this conclusion, given that the model cannot model impact craters and wave breaking, as stated in the manuscript on L365: "This lack also implies that the model cannot consider the impact crater as long as the steepness of the water surface is not steeper than sub-vertical". I do not think many scientists and engineers would find the shallow-water equations (SWEs) a wise choice to model subaerial landslide-tsunami generation.

- Authors: Context have been added (L423-424) and the sentence is more nuanced (L425).

L402 and Fig. 10: "…by the fact that the results of our model also fit well with those experiments." I understand that this mainly based on the comparison in Fig. 10. Yes, the data scatter between the +/-30% for P < 9 (investigated in the original study). However, it is obvious that the data trend systematic deviates from the prediction, for both P < 9 and more significantly over the full presented range of P. It is further obvious that the trend of the data conducted for water depths of up to 0.17 m is very different for the experiments conducted at water depths of 0.10, 0.08 and 0.05 m. There are clear recommendations in the technical literature that laboratory experiments should not be performed certain Weber and Reynolds number limitations (roughly corresponding to water depths smaller than 0.20 m) to avoid significant scale effects. There appears currently to be no attempt in the manuscript to explain this systematic deviation in Fig. 10 with scale effects, or another potential explanation offered by the Authors.

- Authors: The results have received more details (L383-386) and the discussion has been extended (L406-409; 431).

L403: "Finally, our model is validated by a benchmark test performed herein, as this approach is very simple to implement and is very efficient in terms of computational resources. Therefore, we consider our model as a tool of choice for the assessment of landslide-generated tsunami hazards." I agree that the model is very efficient, but I do not feel that "Therefore" is justified as the efficiency of a tool is not the only criterion for hazard assessment (it also needs to represent the underlying physics to an appropriate level, as appreciated by the Authors in the article). I do not think it would be a wise choice to select a SWEs model for landslide-tsunami hazard assessment due to points L396 and L402 above and the fact that the model is not able to consider frequency dispersion (partially responsible for the dangerous underestimation in Fig. 9 in some of the panels). I do not think the answer from the Authors in the response "The comments of both reviewers show that they understand the statement of the balance "correctness-efficiency" and that we had to make a choice of the level of approximation. We chose the non-linear shallow water equations (without multiple add-ons), hence, the inherent approximations and incomplete physics (such as frequency dispersion). We won't discuss in the paper the well-known limitations of this approach." is satisfactory. There are efficient, more appropriate and widely applied alternatives to the SWEs (non-hydrostatic non-linear SWEs, RANS equations, coupled approaches, GPU acceleration, etc.) to model landslide-tsunamis available. The Reviewers review the article not for themselves, but on behave of the readership of the Journal. My feeling as a Reviewer is that 95% of the readership of the Journal are not fully aware of the "correctness-efficiency" aspects and the fundamental limitations of the SWEs to model subaerial landslide-tsunami generation and propagation and will be misled by such statements if the limitations are not better communicated in the article.

- Authors: We agree with the reviewer statement and we have developed a more detailed explanation and provided a better context (L45-48; 57-59; 255-257; 349-351; 431; 433-435)

**Suggested grammatical corrections and minor points:**

- Authors: All the suggested corrections have been performed

Abstract: There are still formatting aspects which should be improved, e.g. the abstract should be presented as 1 paragraph (not several small ones) and the text/figures need to be arranged such that there are no large free spaces on the pages (see e.g. at the bottom of page 10 or 13).

L57: Please add the missing free space in "whichis".

L64: Please drop one of the "that".

L84: Please revise the unclear expression "static critical state friction".

L90: Please replace "paper" (rather informal) with "article".

L96: I do not think "eq. 1" is the common writing style in this journal, normally it would be "(1)". Please check and apply it correctly for all equations.

L100: Again, it is common practice to write parameters in italic in research articles. This is still a general issue in the article, there are many parameters with an inconsistent writing style (L100, L106, L147, L218, caption Fig. 2, L252, L254, L255, L256, L257, L279, captions of Figs. 5, 7, 8, 9 and 10 and many times in the Notation).

L121 (Eq. (8)): On the other hand, numbers, "sin", "cos" and "tan", Fr, P, Re and brackets should not be written in italic (Eqs. (8), (9), (10), (11) and (12), L147, L153, L155, Eqs. (20), (21) and (22), Table 1, L248, L329, L331, L335, L336, L338, L341, Eq. (37), L345, Eq. (39), L350, L355, L356, L400 and in the Notations).

L181: Please write "…when applied in our, never fit the experimental data."

Legend of Fig. 2: Please improve the presentation and remove the typo in "Differnce"

Figs. 4, 6, 7 and 8: Should it read "z" and "x" rather than "Z" and "X" on the axes? Further, some of the numbers on the axes are too small.

L212-L218 and legend of Fig. 2: The writing style of the shape factor is inconsistent (SF, FS, S_F (F as subscript)).

L231: Please write "…in two-dimensions."

L235: Please write "…the momentum transfer is… the wave propagates…"

L257 and L258: A full stop after "al" is required.

L266: The meaning of the word "discrimination" in this context is unclear.

L272: Please write "…identify the best…"

L284: Please write "…is the finally chosen rheological model."

L339: The parameters have just been introduced, so just write "The relationships between P and Fr and S…"

L370: Please write "…lack of modelling frequency dispersion…".

Notation: P is used for two different parameters. Also S is used twice, this should not be the case.

References: Please remove inconsistencies such as inconsistent use of upper and lower case letters in the article titles, abbreviation and no abbreviation of Journal titles, typos (e.g. L437 "genrated"), etc. This was already pointed out in the 1st review round.